# Glycosylphosphatidylinositol biosynthesis and remodeling are required for neural tube closure, heart development, and cranial neural crest cell survival

Marshall Lukacs[1,2], Tia Roberts[1], Praneet Chatuverdi[3], Rolf W Stottmann[1,2,3,4]*

[1]Division of Human Genetics, Cincinnati Children's Medical Center, Cincinnati, United States; [2]Medical Scientist Training Program, Cincinnati Children's Medical Center, Cincinnati, United States; [3]Division of Developmental Biology, Cincinnati Children's Medical Center, Cincinnati, United States; [4]Department of Pediatrics, University of Cincinnati, Cincinnati, United States

**Abstract** Glycosylphosphatidylinositol (GPI) anchors attach nearly 150 proteins to the cell membrane. Patients with pathogenic variants in GPI biosynthesis genes develop diverse phenotypes including seizures, dysmorphic facial features and cleft palate through an unknown mechanism. We identified a novel mouse mutant (*cleft lip/palate, edema and exencephaly; Clpex*) with a hypo-morphic mutation in *Post-Glycophosphatidylinositol Attachment to Proteins-2 (Pgap2)*, a component of the GPI biosynthesis pathway. The *Clpex* mutation decreases surface GPI expression. Surprisingly, *Pgap2* showed tissue-specific expression with enrichment in the brain and face. We found the *Clpex* phenotype is due to apoptosis of neural crest cells (NCCs) and the cranial neuroepithelium. We showed folinic acid supplementation *in utero* can partially rescue the cleft lip phenotype. Finally, we generated a novel mouse model of NCC-specific total GPI deficiency. These mutants developed median cleft lip and palate demonstrating a previously undocumented cell autonomous role for GPI biosynthesis in NCC development.
DOI: https://doi.org/10.7554/eLife.45248.001

*For correspondence: rolf.stottmann@cchmc.org

**Competing interests:** The authors declare that no competing interests exist.

## Introduction

Inherited glycophosphatidylinositol deficiency (IGD) disorders are a class of congenital disorders of glycosylation that affect the biosynthesis of the glycosylphosphatidylinositol (GPI) anchor. The clinical spectrum of IGDs is broad and includes epilepsy, developmental delay, structural brain malformations, cleft lip/palate, skeletal hypoplasias, deafness, ophthalmological abnormalities, gastrointestinal defects, genitourinary defects, heart defects, Hirschsprung's disease, hyperphosphatasia, and nephrogenic defects (*Kinoshita, 2014*; *Knaus et al., 2018*; *Tarailo-Graovac et al., 2015*; *Bellai-Dussault et al., 2019*). However, across all GPI deficiency disorders, the most penetrant defects affect the central nervous system and the craniofacial complex (*Kinoshita, 2014*; *Knaus et al., 2018*; *Tarailo-Graovac et al., 2015*; *Bellai-Dussault et al., 2019*). Indeed, automated image analysis was able to predict the IGD gene mutated in each patient from facial gestalt (*Knaus et al., 2018*). Interestingly, facial gestalt was a better predictor of patient mutation than analysis of the degree of GPI biosynthesis by flow cytometry. Little is known about the mechanism(s) that causes these phenotypes or why disparate tissues are differentially affected. We sought to determine the mechanism responsible for these phenotypes using a novel mouse model of reduced enzymatic function within the GPI biosynthesis pathway.

**eLife digest** Many of the proteins that cells produce have sugar molecules attached to them. These additions, called glycosylations, often help to deliver proteins to the parts of the cell where they are needed. In some genetic disorders, individuals have gene mutations that prevent glycosylation from occurring properly. This can lead to a variety of symptoms including seizures, cleft palates and heart defects. It was not clear how changes in glycosylation cause these symptoms.

A GPI anchor is a specific glycosylation that helps to attach many different proteins to the outer membrane of cells. Lukacs et al. created mouse models with genetic mutations that prevent GPI anchors from forming correctly, and studied the effects these had in mouse embryos. This revealed that a loss of GPI anchors early in embryonic development causes the cells that produce the face to die, as they are very sensitive to an early loss of glycosylation. Because too few face cells survive, embryos develop cleft palate, and other reductions in facial tissues. However, giving the embryos supplements of folinic acid in the womb reduced these effects.

In the future, further experiments using the genetically altered mice generated by Lukacs et al. could explore how glycosylation affects the development of other tissues and organs, like the heart and liver. This could ultimately help researchers to predict the effects of certain genetic conditions and to develop new treatments for them.

DOI: https://doi.org/10.7554/eLife.45248.002

The GPI anchor is a glycolipid added post-translationally to nearly 150 proteins which anchors them to the outer leaflet of the plasma membrane and traffics them to lipid rafts (*Kinoshita, 2014*). The biosynthesis and remodeling of the GPI anchor is extensive and requires nearly 30 genes (*Kinoshita, 2014*). Once the glycolipid is formed and transferred to the C-terminus of the target protein by a variety of Phosphatidylinositol Glycan Biosynthesis Class (PIG proteins), it is transferred to the Golgi Apparatus for remodeling by Post-GPI Attachment to Proteins (PGAP proteins). One of these PGAP proteins involved in remodeling the GPI anchor is Post-Glycosylphosphatidylinositol Attachment to Proteins 2 (PGAP2). PGAP2 is a transmembrane protein that catalyzes the addition of stearic acid to the lipid portion of the GPI anchor and cells deficient in *Pgap2* lack stable surface expression of a variety of GPI-anchored proteins (GPI-APs) (*Kinoshita, 2014*; *Hansen et al., 2013*). Autosomal recessive mutations in *PGAP2* cause Hyperphosphatasia with Mental Retardation 3 (HPMRS3 OMIM # 614207), an IGD that presents with variably penetrant hyperphosphatasia, developmental delay, seizures, microcephaly, heart defects, and a variety of neurocristopathies including Hirschsprung's disease, cleft lip, cleft palate, and facial dysmorphia (*Hansen et al., 2013*; *Jezela-Stanek et al., 2016*; *Krawitz et al., 2013*; *Naseer et al., 2016*). Currently, there is no known molecular mechanism to explain the cause of these phenotypes or therapies for these patients.

In a forward genetic ENU mutagenesis screen, we previously identified the *Clpex* mouse mutant with Cleft Lip, Cleft Palate, Edema, and Exencephaly (*Clpex*) (*Stottmann et al., 2011*). Here, we present evidence that this mutant phenotype is caused by a hypo-morphic allele of *Pgap2.* To date, embryonic phenotypes of GPI biosynthesis mutants have been difficult to study due to the early lethal phenotypes associated with germline knockout of GPI biosynthesis genes (*McKean and Niswander, 2012*; *Nozaki et al., 1999*; *Mohun et al., 2013*; *Zoltewicz et al., 1999*). In this study, we took advantage of the *Clpex* hypo-morphic mutant and a conditional knockout of GPI biosynthesis to determine the mechanism of the various phenotypes and tested the hypothesis that GPI-anchored Folate Receptor 1 (FOLR1) is responsible for the phenotypes observed.

## Results

### The *Clpex* mutant phenotype is caused by a missense mutation in *Pgap2*

We previously identified the *Clpex (cleft lip and palate, edema, and exencephaly)* mutant in a mouse N-ethyl-N-nitrosourea (ENU) mutagenesis screen for recessive alleles leading to organogenesis phenotypes (*Stottmann et al., 2011*). *Clpex* homozygous mutants displayed multiple partially penetrant phenotypes. In a subset of 70 mutants from late organogenesis stages (~E16.5-E18.5), we noted

cranial neural tube defects (exencephaly) in 61 (87%), cleft lip in 22 (31%), cleft palate in 13 (19%), and edema in six embryos (9%) (*Figure 1A–H*). Skeletal preparations of *Clpex* mutants identified a defect in frontal bone ossification (*Figure 1I–L*, n = 5/5 mutants) and a statistically significant decrease in limb length (*Figure 1M–P*). We previously reported a genetic mapping strategy with the Mouse Universal Genotyping Array which identified a 44 Mb region of homozygosity for the mutagenized A/J genome on chromosome 7 (*Figure 1Q*) (*Stottmann et al., 2011*). We then took a whole exome sequencing approach and sequenced 3 *Clpex* homozygous mutants. Analysis of single base pair variants which were homozygous in all three mutants with predicted high impact as determined by the sequence analysis pipeline (e.g. missense variants in conserved residue, premature stop codons, etc.) and not already known strain polymorphisms in dbSNP left only one candidate variant (*Table 1*). This was a homozygous missense mutation in the initiating methionine (c.A1G, p.M1V) in exon 3 of *post-GPI attachment to proteins 2* (*Pgap2*). We confirmed the whole exome sequencing result by Sanger Sequencing (*Figure 1R*). This mutation abolishes the canonical translation start codon for *Pgap2*. However, there are multiple alternatively spliced transcripts that may lead to production of variant forms of *Pgap2*.

To determine whether the *Clpex* phenotype was caused by the missense mutation in *Pgap2*, we performed a genetic complementation test using the $Pgap2^{tm1a(EUCOMM)Wtsi}$ (hereafter referred to as $Pgap2^{null}$) conditional gene trap allele. We crossed $Pgap2^{Clpex/+}$ heterozygotes with $Pgap2^{null/+}$ heterozygotes to generate $Pgap2^{Clpex/null}$ embryos. $Pgap2^{Clpex/null}$ embryos displayed neural tube defects, bilateral cleft lip, and edema similar to the $Pgap2^{Clpex/Clpex}$ embryos at E13.5 (*Figure 2A–H*). $Pgap2^{Clpex/null}$ embryos also displayed micro-opthalmia (*Figure 2F*) and more penetrant cleft lip and edema phenotypes than observed in $Pgap2^{Clpex/Clpex}$ homozygotes (*Figure 2I*). $Pgap2^{Clpex/null}$ embryo viability was decreased with lethality at approximately E13.5-E14.5, precluding analysis of palatal development in these mutants. Histological analysis of the heart in $Pgap2^{Clpex/null}$ E13.5 embryos showed pericardial effusion, a reduction in thickness of the myocardium, and an underdeveloped ventricular septum and valves (*Figure 2J–Q*). As the *Clpex* allele failed to complement a null allele of *Pgap2*, we concluded the *Clpex* phenotype is caused by a hypo-morphic allele of *Pgap2*. $Pgap2^{null/null}$ embryos are resorbed before E9.0 due to early embryonic lethality, whereas $Pgap2^{Clpex/Clpex}$ homozygotes survive to E18.5 confirming the *Clpex* allele is indeed a hypo-morphic allele of *Pgap2* which preserves some function as compared to a true null (*Mohun et al., 2013*).

To determine the possible protein expression of alternatively spliced forms of PGAP2 in the *Clpex* mutant, we performed western immunoblotting with a commercially available antibody against PGAP2 but were unable to detect endogenous PGAP2 in cells or mouse tissues (data not shown). Therefore, we cannot definitively address the presence of alternatively spliced variants of PGAP2 in the *Clpex* mutant. However, a search for *Pgap2* alternatively spliced transcripts identifies 25 alternatively spliced transcripts, 13 of which are protein coding (*Supplementary file 1*) (*Hunt et al., 2018*). While the majority of alternatively spliced transcripts utilize the start codon mutated in the *Clpex* allele, variant *Pgap2-203* utilizes an alternative start codon which significantly alters the C-terminal domain of the protein when compared to the canonical transcript *Pgap2-225* (*Figure 2—figure supplement 1*). The differences in the C-terminal domain between these transcript variants include the cytoplasmic tail and the first helical domain that is predicted to traverse the Golgi membrane. However, the variants are very similar beyond these domains and it is possible the hypo-morphic phenotype of *Clpex* mutants compared to $Pgap2^{null/null}$ mutants may be due to some residual function of transcript variant *Pgap2-203*.

## *Pgap2* is dynamically expressed throughout development

Based on the tissues affected in the *Clpex* mutant, we hypothesized *Pgap2* is expressed in the neural folds and facial primordia of the developing mouse embryo at early stages. We used the lacZ expression cassette within the $Pgap2^{null}$ allele to perform detailed expression analysis of *Pgap2* throughout development (n = 21 litters at multiple developmental stages). We performed RNA in situ hybridization in parallel for some stages to test the fidelity of the lacZ expression and found high concordance. *Pgap2* was expressed relatively uniformly and ubiquitously at neurulation stages in the mouse from E7.5-E8.5 (*Figure 3A–C*). We also noted extraembryonic expression at E7.5, consistent with the abnormal placental development observed in $Pgap2^{null/null}$ embryos (*Figure 3A,B*) (*Mohun et al., 2013*). At E9.0–10.5, there was clear enrichment of *Pgap2* expression in the first branchial arch (*Figure 3D–L*). *Pgap2* RNA in situ hybridization identified a similar pattern of expression

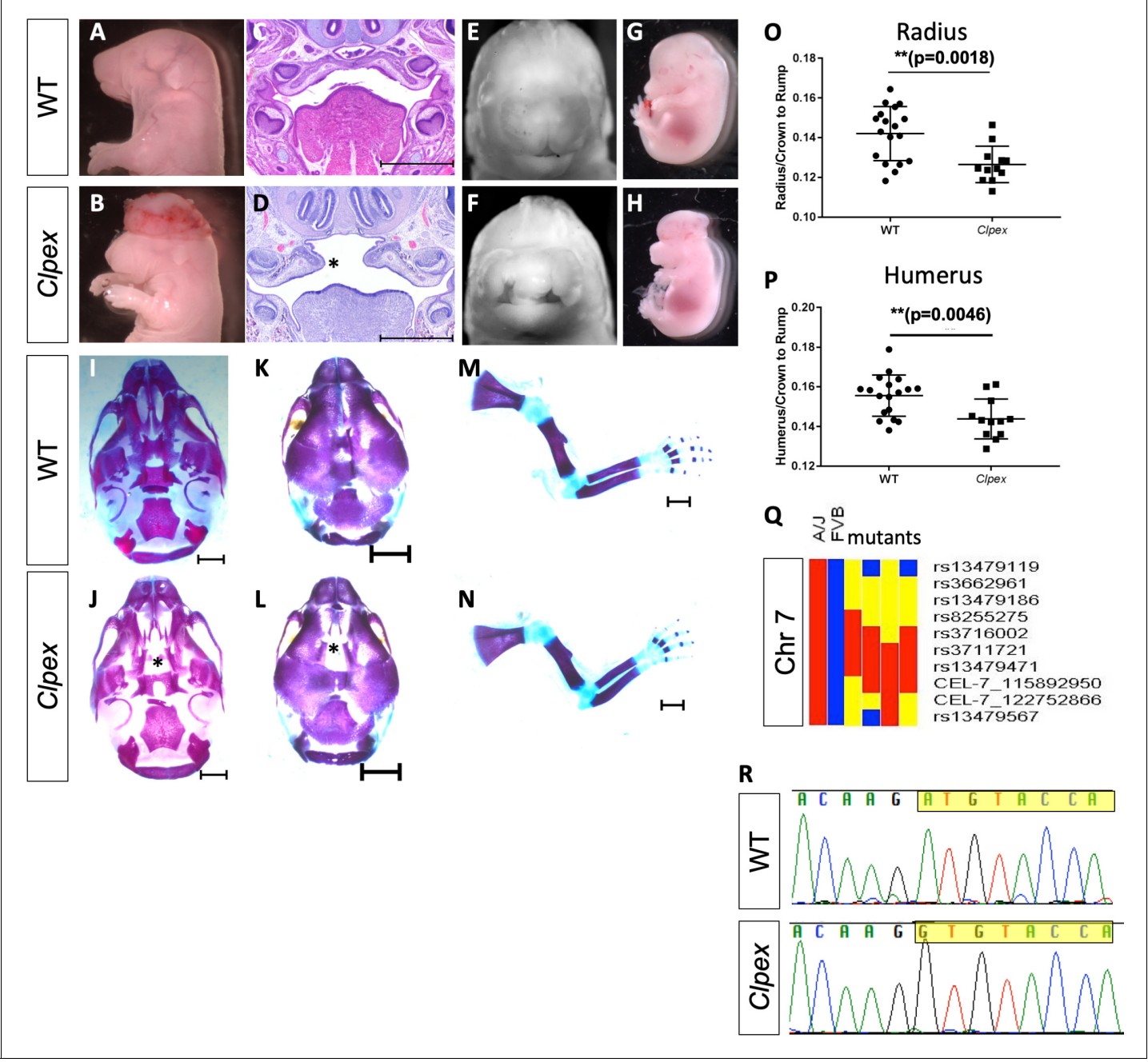

**Figure 1.** The *Clpex* mutant phenotype is caused by a hypo-morphic mutation in *Pgap2*. Whole mount E18.5 (**A,E**) and E15.5 (**G**) WT embryos. Whole mount E18.5 (**B, F**) and E15.5 (**H**) *Clpex* mutant embryos. H&E staining of WT E15.5 (**C**) and *Clpex* (**D**) coronal sections. Skeletal preparation of WT skull ventral view (**I**), dorsal view (**K**). Skeletal preparation of *Clpex* mutant skull ventral view (**J**) and dorsal view (**L**). Asterick indicates absent palatine bone in mutant (**L**). Skeletal preparation of WT limb (**M**), and *Clpex* mutant limb (**N**). Quantification of WT and mutant radial (**O**) and humeral (**P**) length normalized to the crown to rump ratio. Mapping data for *Clpex* mutation (**Q**). Sanger sequencing of *Pgap2* exon three in WT and *Clpex* mutant with exon three highlighted starting at the initiating methionine (**R**). Scale bar indicates 500 μm in C,D and 1 mm in I-N. (**p<0.01).
DOI: https://doi.org/10.7554/eLife.45248.003

The following source data is available for figure 1:

**Source data 1.** Length of radius and humerus in wildtype and Clpex mutants.
DOI: https://doi.org/10.7554/eLife.45248.004

**Table 1.** Exome analysis identifies variant in *Pgap2*.

| Variant filters | Number |
| --- | --- |
| Total variants | 145,956 |
| Homozygous in all three mutants | 120,393 |
| Chromosome 7 | 5854 |
| 81–125 Mb | 2196 |
| 'High' impact | 9 |
| Not in dbSNP | 7 |
| Single base pair change | 1: *Pgap2* |

DOI: https://doi.org/10.7554/eLife.45248.005

as observed in the *Pgap2* LacZ reporter allele at E9.5 (*Figure 3F*). At E9.5 and E10.5, expression was enriched in the limb bud, somites, first branchial arch, eye, forebrain and midbrain (*Figure 3E–L*). There was also increased expression at the medial aspects of both medial and lateral nasal processes at E10.5 (*Figure 3K–L*), and strong expression in the heart starting at E11.5 (*Figure 3N*). At later organogenesis stages, *Pgap2* showed more regionalized and enriched expression, including in the ganglion cell layer of the retina at E11.5 and E14.5 (*Figure 3M*, S2F). At E16.5 *Pgap2* was expressed in the salivary gland, epidermis, stomach, nasal conchae, myocardium, bronchi, kidney, uroepithelium, lung parenchyma, a specific layer of the cortex, and ear (*Figure 3—figure supplement 1*). Interestingly, *Pgap2* showed lower expression in the liver (*Figure 3—figure supplement 1H*) and most of the brain except for a thin layer of the cortex and the choroid plexus at E16.5 and P0 (*Figure 3—figure supplement 1J–M*). We also noted expression in the genital tubercle (*Figure 3—figure supplement 1L*). We conclude *Pgap2* shows tissue specific regions of increased expression which may help to explain why certain tissues such as the craniofacial complex, central nervous system, and heart are preferentially affected in GPI biosynthesis mutants. These data are in contrast to previous reports in which some GPI biosynthesis genes are shown to be ubiquitously and uniformly expressed, including *Pign* in the mouse and *pigu* in zebrafish (*McKean and Niswander, 2012*; *Nakano et al., 2010*). Our *Pgap2* expression is more consistent with the expression of *Pigv* which is enriched in *C. elegans* epidermal tissues (*Budirahardja et al., 2015*).

## *Pgap2* is required for the proper anchoring of GPI-APs, including FOLR1

PGAP2 is the final protein in the GPI biosynthesis pathway and catalyzes the addition of stearic acid to the GPI anchor (*Tashima et al., 2006*). In the absence of *Pgap2*, cells lack a variety of GPI-APs on the cell surface leading to a functional GPI deficiency (*Hansen et al., 2013*; *Jezela-Stanek et al., 2016*; *Krawitz et al., 2013*; *Naseer et al., 2016*; *Tashima et al., 2006*). To determine the effect of the *Clpex* mutation on *Pgap2* function, we performed Fluorescein-labeled proaerolysin (FLAER) flow cytometry staining to quantify the overall amount of the GPI anchor on the cell surface. FLAER is a bacterial toxin conjugated to fluorescein that binds directly to the GPI anchor in the plasma membrane (*Brodsky et al., 2000*; *Sutherland et al., 2007*). We hypothesized *Pgap2* function is impaired in *Clpex* mutants due to the ENU mutation in the initiating methionine. We found mouse embryonic fibroblasts (MEFs) from *Clpex* mutants displayed a significantly decreased FLAER staining compared to wildtype MEFs, consistent with a defect in GPI biosynthesis (n = 3 separate experiments with 4 WT and 4 *Clpex* cell lines; *Figure 4A, B*; p=0.0407).

Our genetic complementation analysis results suggested the *Clpex* allele might be a hypo-morphic allele of *Pgap2*. To test this hypothesis, we generated human embryonic kidney (HEK) 293T clones with a 121 bp deletion in exon 3 of *PGAP2* with CRISPR/Cas9 (termed *PGAP^null/null^* cells; *Figure 4—figure supplement 1*). In parallel, we recapitulated the *Clpex* mutation in three independent clones of HEK293T cells by CRISPR/Cas9 mediated homologous directed repair (termed *Clpex KI* Clones 1, 4, and 7; *Figure 4—figure supplement 1*). We found there was a statistically significant decrease in FLAER staining between WT and 3/3 KI clones and the *PGAP2^null/null^* cells. However, we observed a smaller difference in FLAER staining in *PGAP2^null/null^* cells when compared to *Clpex KI* cells (*Figure 4C, D*). Therefore, we conclude the *Clpex* missense mutation severely affects PGAP2

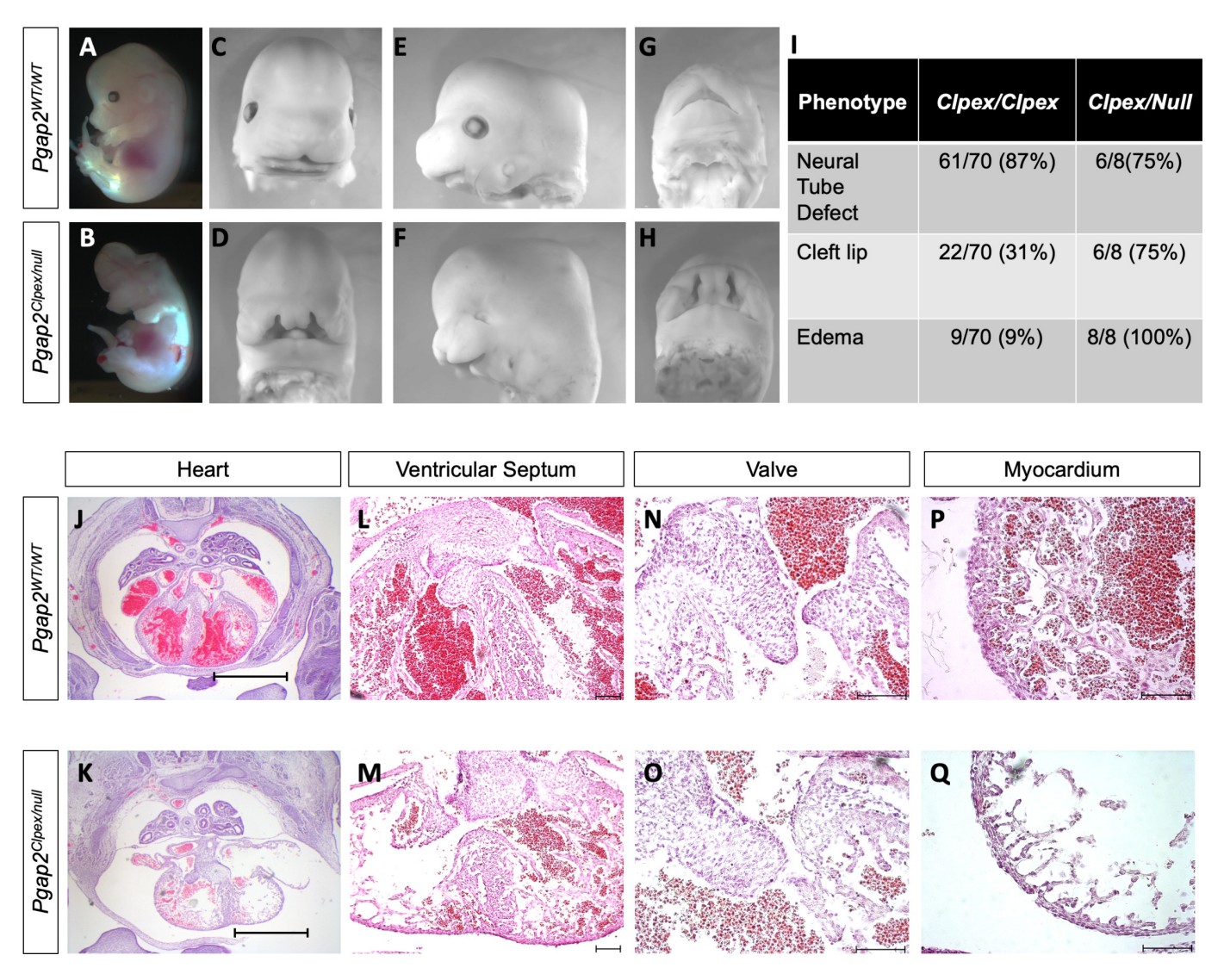

**Figure 2.** *Pgap2^null^* allele fails to complement *Pgap2^Clpex^* allele. Whole mount image of E13.5 WT (**A,C,E,G**) and *Pgap2^Clpex/null^* mutant (**B,D,F,H**). Penetrance of some key phenotypes is compared in I. Cardiac histology of E14.5 WT (**J**) and *Pgap2^Clpex/null^* mutant (**K,**), scale bar indicates 1 mm. Higher power images of the ventricular septum (**L, M**), valve (**N, O**), and myocardial wall (**P, Q**). Scale bar indicates 100 µM in L-Q.
DOI: https://doi.org/10.7554/eLife.45248.006

The following figure supplement is available for figure 2:

**Figure supplement 1.** *Pgap2* alternative transcripts.
DOI: https://doi.org/10.7554/eLife.45248.007

function similar to the effect seen upon total depletion of *PGAP2*. Our *in vivo* findings suggest the *Clpex* mutation produces a hypo-morphic allele of *Pgap2* but our *in vitro* FLAER staining shows functional equivalence between *PGAP2^null/null^* cells and the *Clpex* KI cells. This may reflect subtle differences in the function of *PGAP2 in vitro* versus *in vivo* or reflect a difference between mouse and human *PGAP2* (FLAER was performed in human HEK293T cells). As a positive control, we used CRISPR/Cas9 to delete *phosphatidylinositol glycan anchor biosynthesis, class A* (*Piga; Figure 4—figure supplement 1*). *Piga* is the first gene in the GPI biosynthesis pathway and is absolutely required for GPI biosynthesis (*Jezela-Stanek et al., 2016; Johnston et al., 2012*). We utilized CRISPR/Cas9 to generate a 29 bp deletion in exon 3 of *PIGA*. While not statistically significant, these *PIGA^null/null^*

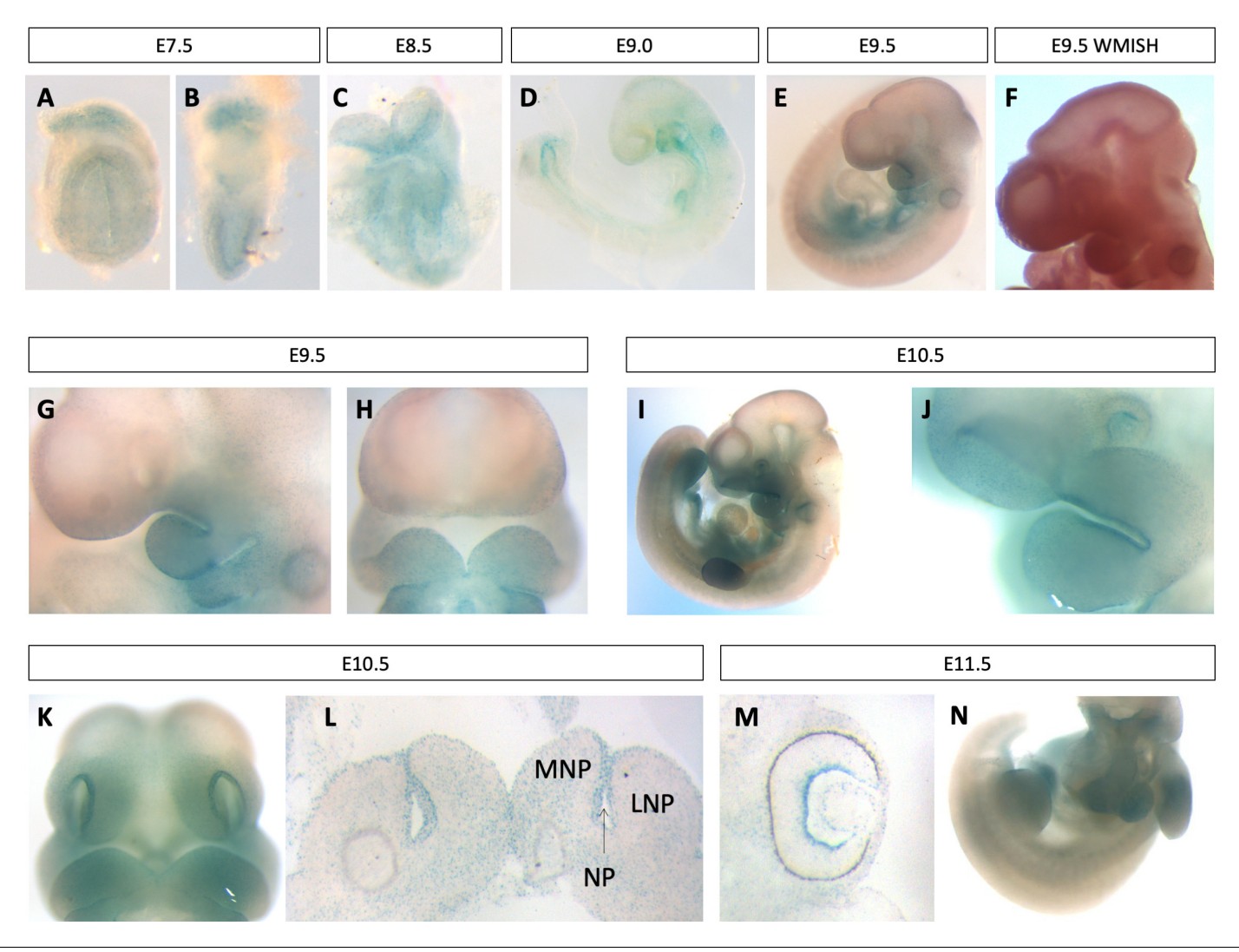

**Figure 3.** *Pgap2* is expressed in neural and craniofacial tissues during development. Whole mount *Pgap2* Xgal staining in E7.5 (**A–B**), E8.5 (**C**), E9.0 (**D**) E9.5 (**E, G, H**), E10.5 (**I–L**), and E11.5 (**M,N**). *Pgap2* RNA in situ hybridization at E9.5 (**F**). Transverse section through the lip at E10.5 (**L**) at the future site of lip closure. Expression is seen in the ganglion cell layer of the retina at E11.5 (**M**). LNP = lateral nasal process, MNP = medial nasal process, NP = nasal pit.

DOI: https://doi.org/10.7554/eLife.45248.008

The following figure supplement is available for figure 3:

**Figure supplement 1.** *Pgap2* expression at later embryonic and early postnatal stages.

DOI: https://doi.org/10.7554/eLife.45248.009

cells trend toward a further decrease in FLAER staining compared to *PGAP2^{null/null}* cells, confirming our staining accurately reflects GPI anchor levels (n = 4 separate experiments; **Figure 4C, D**).

Current estimates suggest nearly 150 genes encode proteins which are GPI anchored (**UniProt Consortium, 2018**). Our manual review of the MGI database found 102 GPI-APs have been genetically manipulated and phenotyped in mice (**Smith et al., 2018**). Of these, the null allele of *Folr1* has a phenotype most similar to the *Clpex* mutant with cranial neural tube defects, cleft lip/palate, and heart outflow tract phenotypes (**Piedrahita et al., 1999**). Tashima et. al. previously showed *PGAP2* is required for stable cell surface expression of FOLR1 in CHO cells (**Tashima et al., 2006**). To confirm this finding, we overexpressed a myc-tagged FOLR1 construct in WT and *PGAP2-^{null/null}* 293 T cells and assessed the presentation on the plasma membrane as marked by immunocytochemistry for wheat germ agglutinin (WGA). We observed a decrease in co-localization of FOLR1

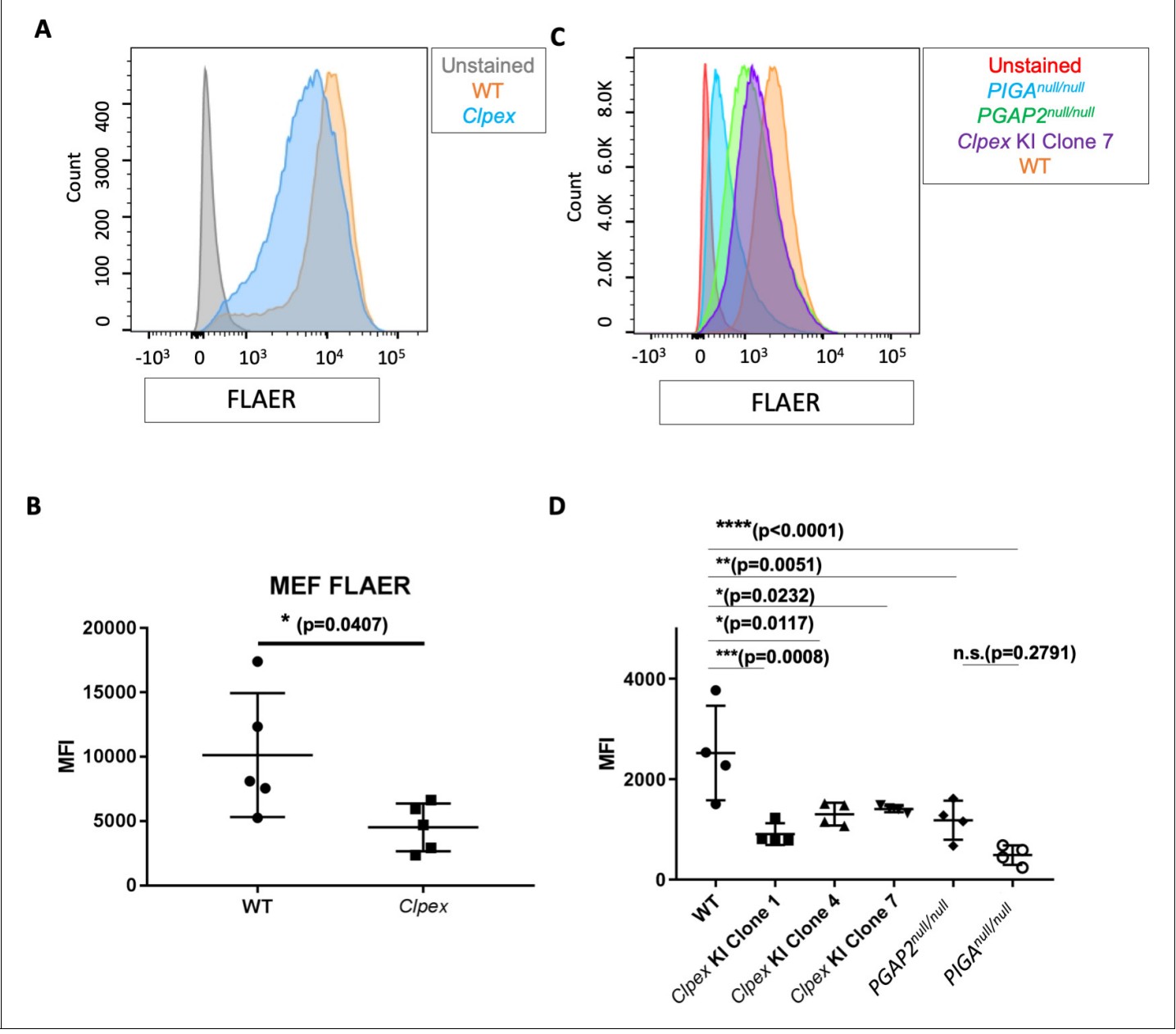

**Figure 4.** *Pgap2* is required for proper anchoring of GPI-APs. (**A**) FLAER staining of WT (orange) and *Clpex* (blue) MEFs, (unstained control in gray) with quantification of Mean Fluorescence Intensity (MFI) (**B**). FLAER staining of WT (orange) *Clpex* KI Clone 7 (purple), $PGAP2^{-/-}$ (green), $PIGA^{-/-}$ (blue) HEK293T cells, and unstained control (red) (**C**) with quantification of MFI (**D**). *p<0.05, **p<0.01, ***p<0.001, ****p<0.0001.

DOI: https://doi.org/10.7554/eLife.45248.010

The following source data and figure supplement are available for figure 4:

**Source data 1.** FLAER Staining (MFI) of MEFs and HEK clones.
DOI: https://doi.org/10.7554/eLife.45248.012

**Figure supplement 1.** Sequencing of CRISPR/Cas9 generated $PIGA^{null/null}$, $PGAP2^{null/null}$, and *Clpex* KI 293T clones.
DOI: https://doi.org/10.7554/eLife.45248.011

with WGA in $PGAP2^{null/null}$ cells compared to controls (*Figure 5A–C,G–I*). In the absence of *PIGA*, cells lack the surface expression of any GPI-APs (*Kinoshita, 2014*). We found $PIGA^{null/null}$ cells showed decreased co-localization of FOLR1 with WGA to a greater degree than that observed in $PGAP2^{null/null}$ cells (representative images from n = 3 technical replicates, *Figure 5D–F*, p<0.0001).

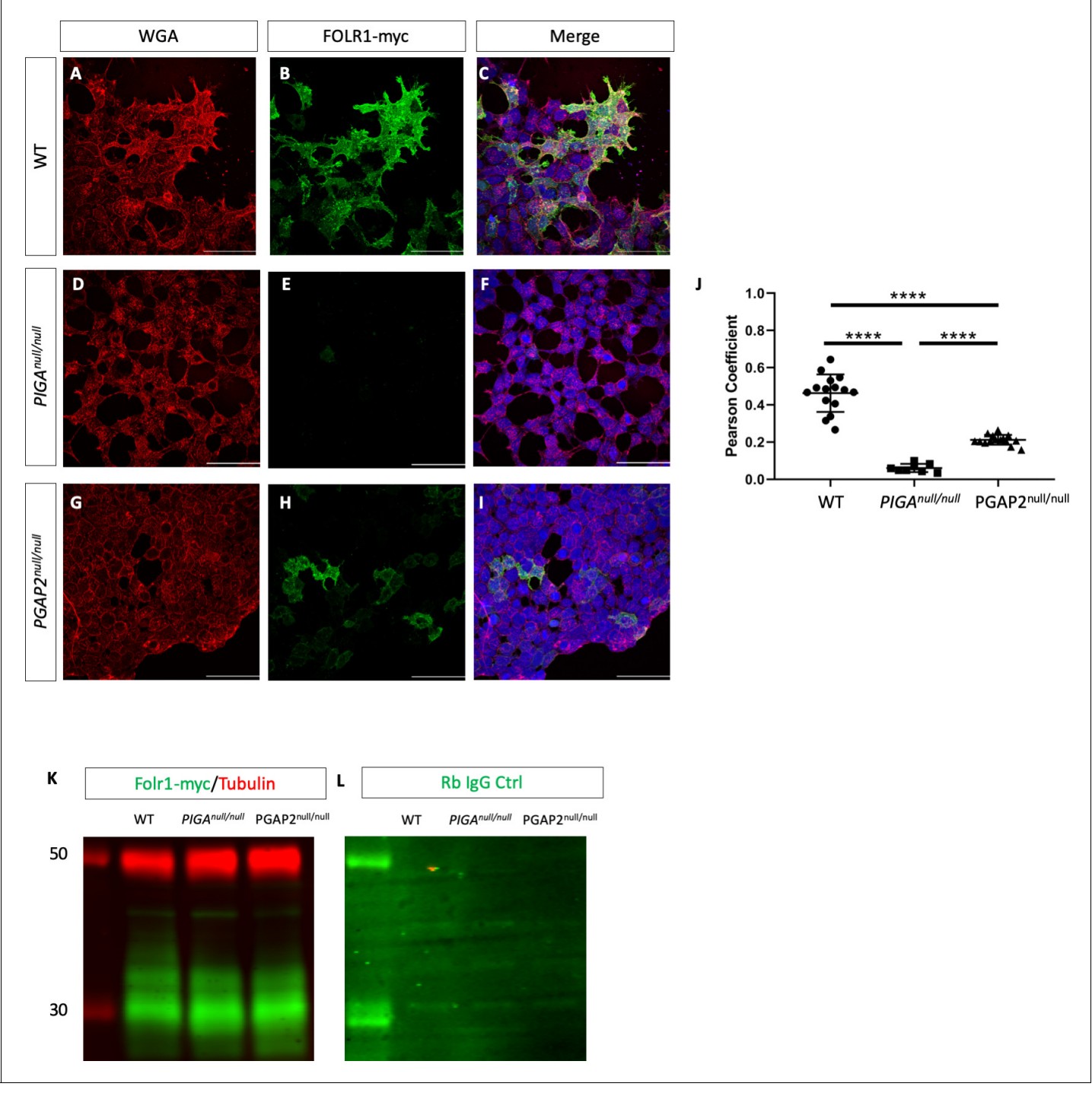

**Figure 5.** Trafficking of FOLR1 to the cell membrane requires GPI biosynthesis and remodeling. Wheat germ agglutinin (WGA) staining in WT (**A**), *PIGA<sup>null/null</sup>* (**D**), *PGAP2<sup>null/null</sup>* (**G**) HEK293T cells. FOLR1-myc staining in WT (**B**), *PIGA<sup>null/null</sup>* (**E**), *PGAP2<sup>null/null</sup>* (**H**) HEK293T cells. Merge of WGA and FOLR1 for WT (**C**), *PIGA<sup>null/null</sup>* (**F**), and *PGAP2<sup>null/null</sup>* (**I**). Pearson Coefficient for co-localization of WGA and FOLR1-myc (**J**). Western blot for αmyc-FOLR1 (green) and αTubulin (red) loading control from cell lysates of WT, *PIGA<sup>null/null</sup>*, and *PGAP2<sup>null/null</sup>* cells overexpressing N-myc tagged FOLR1 (**K**) and Rabbit IgG control for the same cell lysate (**L**). ****p<0.0001, Scale bar indicates 100 μM.

DOI: https://doi.org/10.7554/eLife.45248.013

The following source data is available for figure 5:

**Source data 1.** Quantification of co-localization of WGA and Folr1-myc staining.
DOI: https://doi.org/10.7554/eLife.45248.014

We found $PGAP2^{null/null}$ cells show a severe defect in FOLR1-myc/WGA co-localization but retain more surface staining than $PIGA^{null/null}$ cells (*Figure 5J*, p<0.0001). Therefore, we concluded $PGAP2^{null/null}$ cells have an intermediate defect in FOLR1 membrane trafficking as compared to $PIGA^{null/null}$ cells. However, both $PIGA^{null/null}$ and $PGAP2^{null/null}$ cells produced similar amounts of FOLR1 protein by western blot, indicating that the defect is in membrane trafficking, and not protein production (*Figure 5K,L*).

## Neural crest cells and cranial neuroepithelium display increased apoptosis in the *Clpex* mutant

A number of GPI-APs are critical for cranial neural crest cell (cNCC) migration and survival; including ephrins and FOLR1 (*Santiago and Erickson, 2002*; *Holmberg et al., 2000*; *Li et al., 2011*; *Rosenquist, 2013*; *Rosenquist et al., 2010*; *Spiegelstein et al., 2004*; *Tang et al., 2004*; *Zhu et al., 2007*; *Wahl et al., 2015*). This led us to the hypothesis that cNCC migration may be impaired in *Clpex* mutants ultimately causing the cleft lip and palate phenotype (*Holmberg et al., 2000*). To test whether cNCC migration was impaired in the *Clpex* mutant, we performed a NCC lineage trace using the *Wnt1*-Cre mouse (B6.Cg-$H2afv^{Tg(Wnt1-cre)11Rth}$Tg(Wnt1-GAL4)11Rth/J) in combination with the R26R LacZ reporter (B6.129S4-$Gt(ROSA)26Sor^{tm1Sor}$/J (R26R$^{Tg}$) to create *Wnt1-Cre;R26R$^{Tg/wt}$; Pgap2$^{clpex/clpex}$* mutants in which the NCCs are indelibly labeled with LacZ expression at E9.5 and E11.5 (*Brewer et al., 2004*; *Chai et al., 2000*; *Soriano, 1999*; *Brault et al., 2001*; *Danielian et al., 1998*). We observed no significant deficit in cNCC migration in the mutant embryos as compared to littermate controls at either stage (representative images of n = 2 E9.5 mutants and n = 5 E11.5 mutants; *Figure 6A–F*). However, we observed hypoplasia of the medial and lateral nasal processes at E11.5, suggesting the *Clpex* phenotype is due to earlier defects in NCC survival (*Figure 6E–F*). As *Pgap2* was highly expressed in the epithelium and epithelial barrier defects are a known cause of cleft palate, we next sought to determine whether the epidermis was compromised in the *Clpex* mutant (*Ingraham et al., 2006*). We performed a Toluidine Blue exclusion assay but found no significant defects in barrier formation in the mutant (*Figure 6—figure supplement 1*).

*Folr1$^{-/-}$* mice and zebrafish *Folr1* morphants display increased cell death and decreased proliferation in the facial primordia (*Spiegelstein et al., 2004*; *Zhu et al., 2007*; *Wahl et al., 2015*; *Tang and Finnell, 2003*). We hypothesized a similar mechanism may be responsible for the cleft lip and palate in the *Clpex* mutant embryos. To test this hypothesis, we performed immunohistochemistry for αAP2 to mark NCCs and the apoptosis marker Cleaved Caspase 3 (CC3). We found the cNCCs of the first arch and a specific population of cells within the neuroepithelium were undergoing apoptosis significantly more frequently in *Clpex* homozygous mutants (*Figure 6G–N*). The ratio of CC3-positive to AP2-positive spots revealed a highly significant increase in the percentage of CC3-positive spots in the first arch of *Clpex* mutants (n = 2 or more sections from 3 WT and 6 *Clpex* mutants; *Figure 6Q*; p=0.0045) We also observed apoptosis in the cranial neuroepithelium bilaterally at the dorsolateral hinge points in mutant sections (*Figure 6O,P*). The dorsolateral hingepoints in the anterior neural tube are crucial for proper closure of the neural tube. This apoptosis was exclusively confined to the cranial aspects of the neural tube at the midbrain-hindbrain boundary.

## Dietary folinic acid supplementation partially rescues cleft lip in *Clpex* mutants

Dietary folinic acid supplementation has been shown to rescue the early embryonic lethal phenotype of *Folr1$^{-/-}$* mice and these mice can then survive to adulthood (*Spiegelstein et al., 2004*; *Zhu et al., 2007*; *Tang et al., 2005*). Our data suggest FOLR1 receptor trafficking is impaired in the *Clpex* mutant (*Figure 5*), leading us to hypothesize folinic acid supplementation *in utero* may rescue the *Clpex* phenotype. We further hypothesized the folinic acid diet would have a greater beneficial effect than folic acid as folinic acid (reduced folate) has a higher affinity for other folate receptors including *solute carrier family 19 (folate transporter), member 1 (Slc19a1)* and *solute carrier family 46, member1 (Slc46a1)*, which are not GPI anchored (*Zhao et al., 2001*). In comparison, folic acid has a higher affinity for the GPI anchored folate receptors FOLR1 and FOLR2 (*Jansen and Peters, 2015*; *Jakubowski et al., 2009*). We supplemented pregnant *Clpex* dams from E0-E9.5 or E16.5 with a 25 parts per million (ppm) folinic acid diet, 25 ppm folic acid diet or control diet, and collected *Clpex* mutants for phenotypic analysis and CC3 staining for apoptosis (*Figure 7A*). The

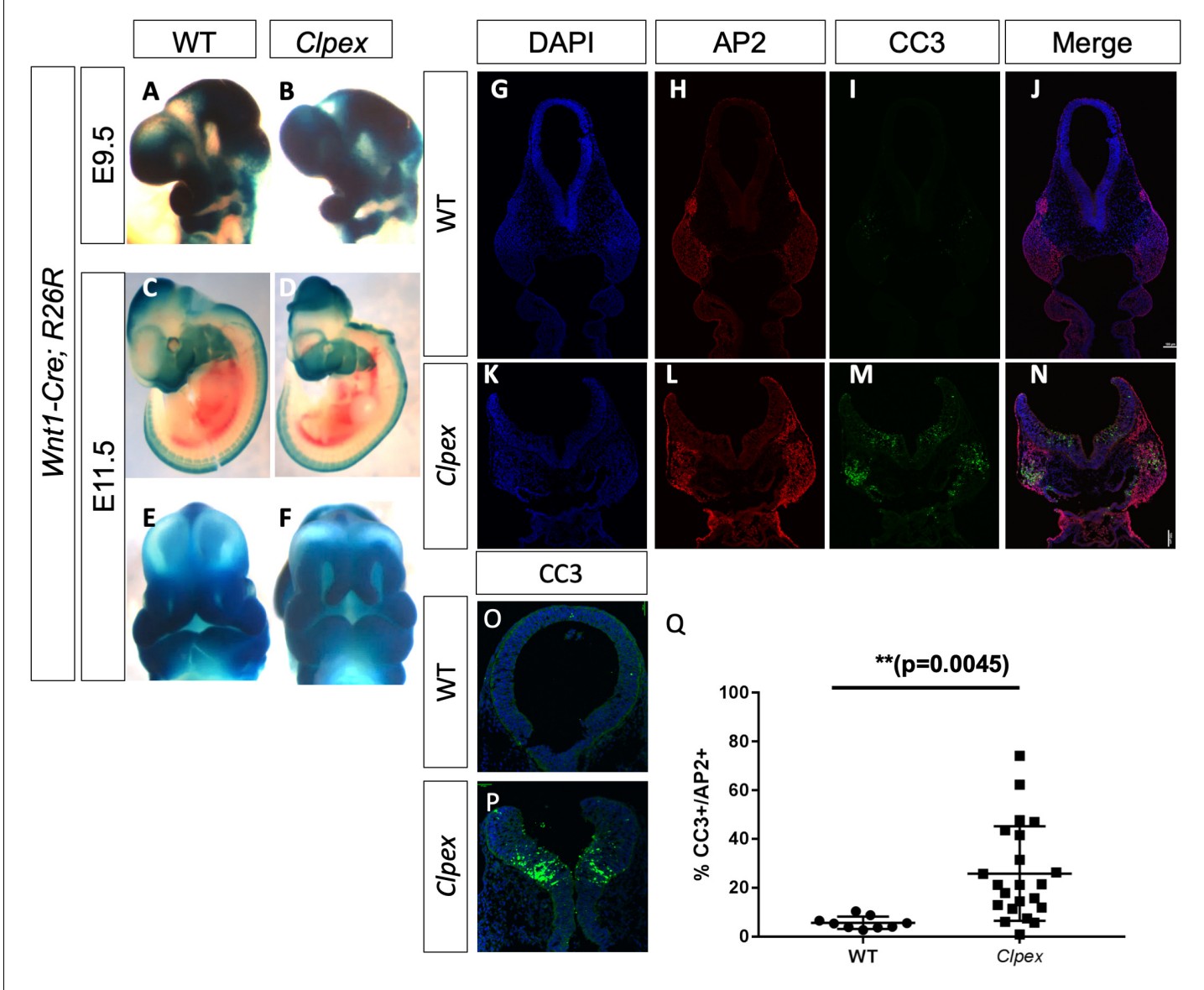

**Figure 6.** *Clpex* cNCCs and neuroepithelium undergo apoptosis at E9.5. *Wnt1*-Cre, R26R NCC lineage trace in WT (**A, C, E**) and *Clpex* mutant (**B,D,F**) at E9.5 (**A,B**) and E11.5 (**C–F**). WT E9.5 embryo stained for DAPI (**G**), AP2 (**H**) CC3 (**I**) and merged image in (**J**). *Clpex* E9.5 embryo stained for DAPI (**K**), AP2 (**L**) CC3 (**M**), and merged image in (**N**). Higher power image of WT (**O**) and *Clpex* mutant (**P**) neuroepithelium stained with CC3 and DAPI. Quantification of CC3+ cells over AP2+ cells in the first branchial arch Region of Interest (**Q**). **p<0.01. Scale bar indicates 100 μm.

DOI: https://doi.org/10.7554/eLife.45248.015

The following source data and figure supplement are available for figure 6:

**Source data 1.** Quantification of CC3-positive and AP2-positive cells in wildtype and mutant.

DOI: https://doi.org/10.7554/eLife.45248.017

**Figure supplement 1.** *Clpex* mutants do not display a defect in barrier formation.

DOI: https://doi.org/10.7554/eLife.45248.016

folinic-acid-treated group had a significantly smaller proportion of mutants with cleft lip (p=0.02), but there was no effect on the incidence of NTD or cleft palate compared to control diet (***Figure 7B, C***; p=0.87). We did note a mild decrease in the number of mutants with edema (***Figure 7B, C***; p=0.06). Consistent with our hypothesis, we found the folinic acid diet reduced the number of mutants with cleft lip by 23% (2/25 mutant vs. 22/70 control), to a greater degree than the 10% reduction observed in folic acid treated embyros (3/14, n.s., ***Figure 7B,C***). Therefore, we

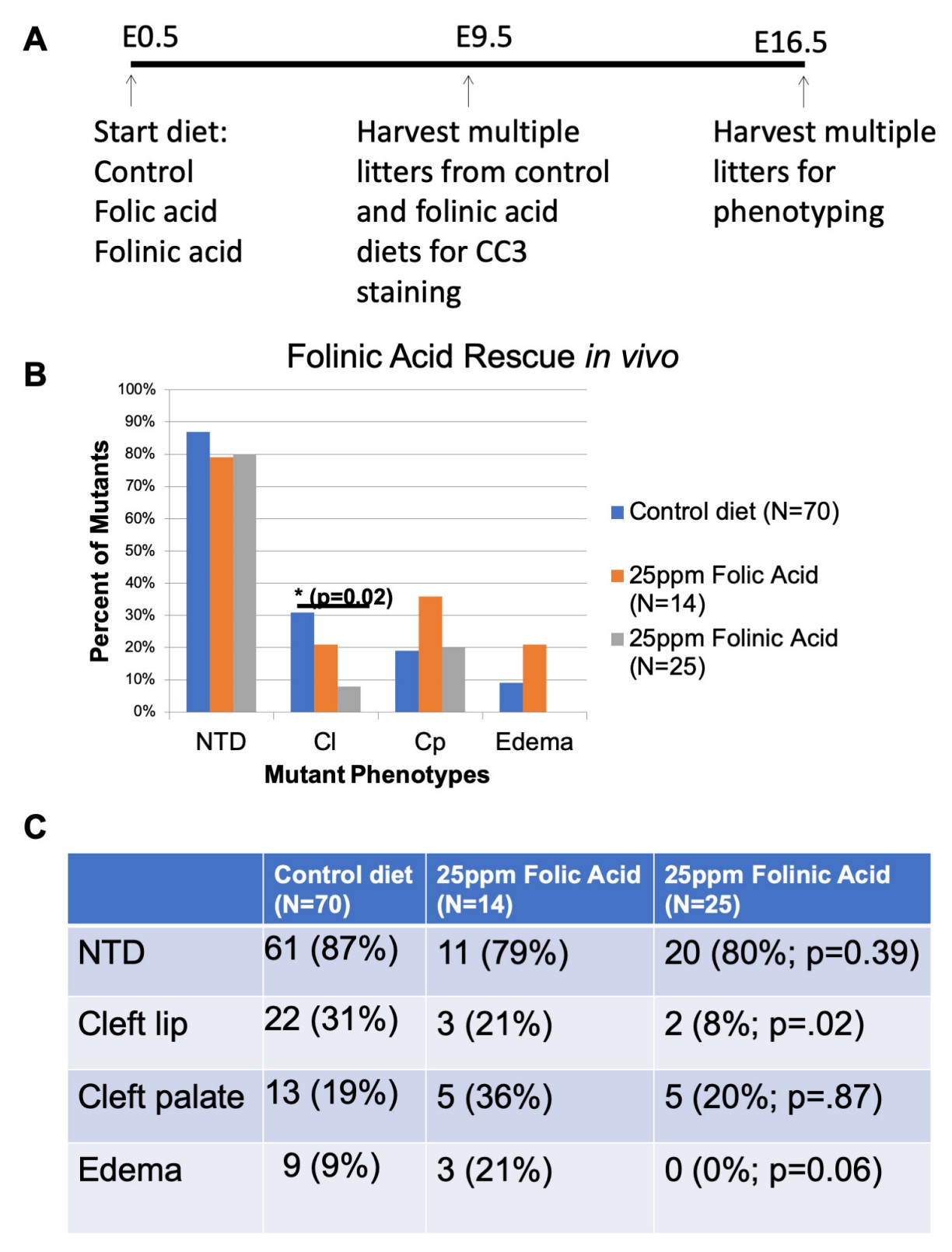

**Figure 7.** Folinic Acid treatment *in utero* partially rescues the cNCC apoptosis and cleft lip in *Clpex* mutants. Schema of diet regimen to evaluate apoptosis and phenotypic rescue in *Clpex* mutants treated with control, 25ppm folic acid, or 25ppm folinic acid diet *in utero* (A). Phenotypes observed in *Clpex* mutants from litters treated from E0-E16.5 with control diet (blue), 25ppm folic acid (orange), or 25ppm folinic acid (gray) (B). Summary of the phenotypes of *Clpex* mutants from litters treated with the indicated diets (C). (*p<0.05).

*Figure 7 continued on next page*

*Figure 7 continued*

DOI: https://doi.org/10.7554/eLife.45248.018

The following figure supplement is available for figure 7:

**Figure supplement 1.** Folinic acid treatment of *Clpex* embyos does not rescue neural crest cell apoptosis.

DOI: https://doi.org/10.7554/eLife.45248.019

conclude folinic acid treatment was more effective than folic acid treatment at reducing the cleft lip phenotype in *Clpex* mutants. We hypothesized this would be via a decrease in NCC apoptosis in folinic acid treated *Clpex* mutants compared to control. However, CC3 staining in control diet treated *Clpex* mutants compared to folinic acid treated *Clpex* mutants showed no statistical difference in the number of apoptotic spots in folinic acid treated embryos at E9.5 (*Figure 7—figure supplement 1*; p=0.3796). These data argue the partial rescue of the cleft lip phenotype in folinic-acid-treated *Clpex* mutants is not due to a decrease in apoptosis at E9.5 in the *Clpex* mutant. This failure of folinic acid treatment to decrease the degree of apoptosis at E9.5 may explain why folinic acid treatment could not rescue the cleft palate or NTD phenotype.

## RNA sequencing reveals changes in patterning genes in *Clpex* mutants

As folinic acid supplementation could not rescue all phenotypes observed in the *Clpex* mutant, we took an unbiased transcriptomic approach to determine the major signaling pathway(s) affected upon reduced *Pgap2* function. RNA sequencing was performed on pooled RNA samples from wild-type and *Clpex* homozygous mutant embryos at E9.5 (5 of each in each RNA pool). Sorting differentially expressed genes in ToppGene showed that the most differentially regulated pathways were sequence-specific DNA binding genes including 39 genes (p=$1.79 \times 10^{-7}$; *Table 2*; *Chen et al., 2009*). Among the sequence-specific DNA binding genes, the majority (23/39 genes in the category) were transcription factors which have been implicated in anterior/posterior (A/P) patterning including *Cdx2, Cdx4,Tbxt, Hmx1, Lhx2,* and *Lhx8* (*Savory et al., 2009*; *Chawengsaksophak et al., 2004*; *van Nes et al., 2006*; *Abe et al., 2000*; *Shedlovsky et al., 1988*; *Munroe et al., 2009*; *Porter et al., 1997*; *Ando et al., 2005*) (*Table 2*). Anterior patterning genes were statistically significantly downregulated and posterior patterning genes were statistically significantly upregulated (*Table 3*). We confirmed changes in expression of three of these A/P patterning defects by RNA in situ hybridization at E9.5 (*Supplementary file 3*). We identified a decrease in *Alx3* in *Clpex* mutants which is both an anterior patterning gene with a prominent role in frontonasal development and a genetic target of folate signaling (*Lakhwani et al., 2010*). We investigated *Lhx8* because it is expressed in the head at E9.5 and *Lhx8*$^{-/-}$ mice develop cleft palate (*Zhao et al., 1999*). We found *Lhx8* was decreased in *Clpex* mutant heads. Finally, the master posterior patterning gene *Tbxt* (*brachyury*) is critical for determining tail length and posterior somite identity (*Abe et al., 2000*; *Shedlovsky et al., 1988*). We found the *Tbxt* expression domain was shifted anteriorly in *Clpex* mutants compared controls at E8.5 (*Supplementary file 3*).

The second and third most altered pathways identified by ToppGene were cholesterol transporter activity (p=$1.1 \times 10^{-6}$), and apolipoprotein binding (p=$2.34 \times 10^{-6}$), respectively. Upon closer inspection, the genes in these categories were largely genes expressed in the mesendoderm including *Alpha fetal protein* and the *Apolipoprotein* gene family. We concluded the decreased expression of these genes in the *Clpex* mutant embryos is consistent with a defect in mesendoderm induction, rather than specific cholesterol and apolipoprotein activities.

Collectively, these findings from our transcriptomic analysis suggest other GPI-Aps involved in A/P patterning and mesendoderm development may be affected in *Clpex* homozygous mutant embryos. Multiple other mutations in GPI biosynthesis genes including *Pgap1* and *Pign* lead to defective CRIPTO mediated NODAL/BMP signaling which affects the formation of the A/P axis in the early gastrulating embryo (*McKean and Niswander, 2012*; *Zoltewicz et al., 1999*; *Chen et al., 2008*; *Zoltewicz et al., 2009*). CRIPTO is a co-receptor for NODAL and necessary for the induction of the anterior visceral endoderm and subsequent forebrain and mesendoderm formation (*Thomas and Beddington, 1996*; *Rossant and Tam, 2009*). Mckean et al found CRIPTO signaling was impaired in the *Pign*$^{Gonzo}$ and *Pgap1*$^{Beaker}$ GPI biosynthesis mutants (*McKean and Niswander, 2012*). Furthermore, stem cells from GPI-deficient clones are unable to respond to TGFβ superfamily

**Table 2.** RNA sequencing ToppGene pathway enrichment analysis.

| GO Category ID | GO Category Name | p-value | q-value Bonferroni | q-value FDR B&H | q-value FDR B&Y | Hit Count in Query List | Hit Count in Genome | Hit in Query List |
|---|---|---|---|---|---|---|---|---|
| GO:0043565 | Sequence-specific DNA binding | 1.79E-07 | 1.53E-04 | 1.53E-04 | 1.12E-03 | 39 | 1096 | CDX2, CDX4, CIART, MYT1, EVX1, BCL11A, NR6A1, LHX8, HMX1, HNF4A, HOXA1, HOXD11, NHLH1, NHLH1, NHLH2, HAND1, TBXT, CREB3L3, NR1H4, PHF21B, TBX15, ALX3, FOXN4, ESX1, POU6F1, EBF1, IFI16, NKX1-2, HEYL, ZSCAN10, NKX2-4, EGR2, NKX2-1, NR2E1, FOXI2, SIX2 |
| GO:0017127 | Cholesterol transporter activity | 1.10E-06 | 9.41E-04 | 4.71E-04 | 3.45E-03 | 5 | 14 | ABCA1, APOA1, APOA2, APOA4 , APOB |
| GO:0034185 | Apolipoprotein binding | 2.34E-06 | 2.01E-03 | 6.68E-04 | 4.90E-03 | 5 | 16 | ABCA1, MAPT, PLG, PCSK9, LIPC |

DOI: https://doi.org/10.7554/eLife.45248.020

members due to defects in GPI anchored co-receptor anchoring (*McKean and Niswander, 2012*; *Chen et al., 2008*). Zoltewicz et al found mutations in *Pgap1* lead to defective A/P patterning by affecting other major signaling pathways including *Wnt* (*Zoltewicz et al., 2009*). Our RNA-Seq results are consistent with the existing literature which has established a critical role for GPI biosynthesis in generating the A/P axis (*Lee et al., 2016*). While this role is well established, few groups have investigated the tissue-specific role of GPI biosynthesis after the A-P axis has been established. Interestingly, two studies have found GPI-APs have cell autonomous roles separately in skin and limb development (*Ahrens et al., 2009*; *Tarutani et al., 1997*). As we found tissue-specific defects in the NCC population in the *Clpex* mutant, we sought to address a larger question and determine the cell autonomous role for GPI-APs in NCC development.

## NCC-specific deletion of *Piga* completely abolishes GPI biosynthesis and leads to median cleft lip, cleft palate, and craniofacial skeletal hypoplasia

We observed cell-type-specific apoptosis in the cNCCs in the *Clpex* mutant and to further our understanding we sought to determine the cell autonomous role of GPI biosynthesis more generally in these cells. *Phosphatidylinositol glycan anchor biosynthesis, class A* (*Piga*) is part of the GPI-N-acetyl-glucosaminyltransferase complex that initiates GPI biosynthesis from phosphatidylinositol and N-acetylglucosamine (*Kinoshita, 2014*). *Piga* is totally required for the biosynthesis of all GPI anchors and *Piga* deletion totally abolishes GPI biosynthesis (*Nozaki et al., 1999*; *Miyata et al., 1993*; *Watanabe et al., 1996*). We first performed RNA whole mount in situ hybridization for *Piga* and showed it has a similar regionalized expression as we observed in the *Pgap2* expression experiments. *Piga* expression at E11.5 is enriched in the first branchial arch, heart, limb, and CNS (representative images from n = 8 antisense and two sense controls over three separate experiments, *Figure 8A–F*). However, *Piga* showed a unique enrichment in the medial aspect of both medial nasal processes as opposed to the *Pgap2* expression which appeared to line the nasal pit epithelium (*Figure 8C*). Other GPI biosynthesis genes showed a similar regionalization pattern of expression (*Figure 8—figure supplement 1*).

To determine the NCC specific role for GPI biosynthesis, we generated a novel model of tissue-specific GPI deficiency in the neural crest cell lineage with *Piga*$^{flox/X}$; *Wnt1*-Cre mosaic conditional KO (cKO) mutants and *Piga*$^{flox/Y}$; *Wnt1*-Cre hemizygous cKO mutants. As *Piga* is located on the X chromosome, females with genotype *Piga*$^{flox/X}$; *Wnt1*-Cre$^{+/-}$ will develop genetic mosaicism due to random X inactivation whereas males with genotype *Piga*$^{flox/Y}$; *Wnt1*-Cre$^{+/-}$ will develop total *Piga* deficiency in the *Wnt1*-Cre lineage. To confirm the loss of GPI biosynthesis in these mutants, we cultured Mouse Palatal and Nasal Mesenchymal Cells (MPNMCs) from WT and mutant palates and performed FLAER staining as above. We found mutant MPMNCs lack virtually all GPI anchors on the

**Table 3.** Anterior/posterior transcription factors differentially expressed in *Clpex* mutants compared to controls.

Table 3A

| Anterior TFs | | | TPM | |
|---|---|---|---|---|
| Gene | Log$_2$ FC | p-value | Wild-type | *Clpex* |
| *Lhx8* | -1.56 | 0.004589 | 1.95 | 0.59 |
| *Alx3* | -0.71 | 0.001724 | 18.68 | 10.36 |
| *Nkx2-4* | -2.37 | 0.009815 | 1.89 | 0.31 |
| *Hmx1* | -1.75 | 3.308E-06 | 3.9 | 1.04 |
| Table 3B | | | | |
| Posterior TFs | | | TPM | |
| Gene | Log$_2$ FC | p-value | Wild-type | *Clpex* |
| *Cdx2* | 1.89 | 0 | 4 | 13.53 |
| *Cdx4* | 2.04 | 0 | 3.21 | 12.12 |
| *Evx1* | 1.60 | 3.14E-08 | 1.06 | 2.97 |
| *Hoxc10* | 1.12 | 8.57E-11 | 10.01 | 19.83 |
| *Hoxd11* | 1.07 | 1.69E-06 | 2.43 | 4.59 |
| *Nkx1-2* | 1.77 | 2.50E-12 | 1.97 | 6.13 |
| *Tbxt* | 1.69 | 0 | 3.35 | 9.89 |

DOI: https://doi.org/10.7554/eLife.45248.021

cell surface (n = 4 WT and three mutant cell lines stained in two separate experiments) (*Figure 8G, H*, p=0.0036).

Analysis of mosaic cKOs at E15.5–16.5 showed mild median cleft lip and cleft palate in all mutants examined (n = 6 mutants; *Figure 9A–F*). Hemizygous cKOs showed a more severe median cleft lip and cleft palate in all mutants examined (n = 7 mutants; *Figure 9G–L*). Skeletal preparations to highlight bone and cartilage demonstrates hypoplasia of the craniofacial skeleton and cleft palate (n = 5/5 mutants examined; *Figure 9M–R*).

These data confirm for the first time a cell autonomous role for GPI biosynthesis in cNCCs during development. As we reduced the amount of *Piga* from the mosaic cKO to the hemizgous cKO, we observed a worsening of the cleft lip/palate phenotype including more severe hypoplasia of the palatal shelves and widening of the median cleft lip. These data are consistent with the hypothesis that the dosage of GPI biosynthesis is related to the severity of the phenotype with mutants with less residual GPI anchor expression showing more severe phenotypes. Surprisingly, we found hemizygous cKO mutants are capable of forming all the bones and cartilage of the craniofacial skeleton, although they are all hypoplastic.

These data are consistent with the hypothesis that GPI biosynthesis is involved in the survival of early cNCCs as we observed in the *Clpex* homozygous mutants and not in the later patterning or differentiation of cNCCs. The critical requirement for GPI biosynthesis appears to be at early stages of cNCC survival just after they have migrated from the dorsal neural tube, and before they have committed to differentiation into bone or cartilage.

## Discussion

In this study, we aimed to determine the role of GPI biosynthesis in craniofacial development with two novel models of GPI deficiency. First, we characterized the phenotype of the ENU-induced *Clpex* allele which shows partially penetrant cranial neural tube defects, bilateral cleft lip/palate, and edema. We found by mapping and whole exome sequencing the *Clpex* mutation is a missense allele in the initiating methionine of *Pgap2*, the final enzyme in the GPI biosynthesis pathway. The *Clpex* allele failed to complement a null allele of *Pgap2*, confirming the *Clpex* mutant is caused by a hypomorphic mutation in *Pgap2*. By expression analysis with a *Pgap2^tm1a (lacZ)* reporter allele, we found *Pgap2* is enriched in the first branchial arch, limb bud, neuroepthelium and around the interior

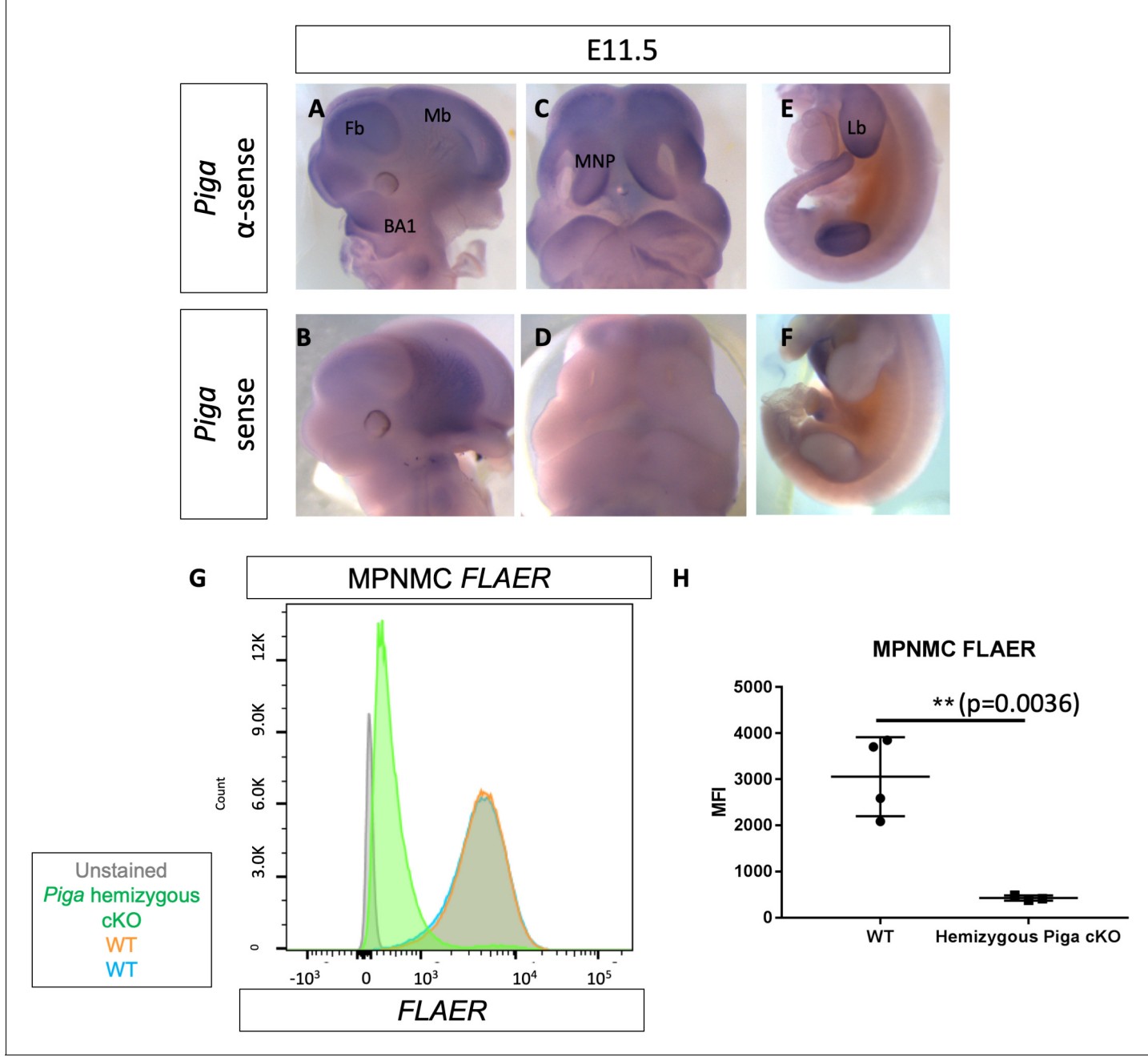

**Figure 8.** *Piga* is expressed in the first branchial arch, medial nasal process, limb bud and deletion of *Piga* in the *Wnt1*-Cre lineage results in NCC cells that lack GPI biosynthesis. WMISH of WT E11.5 embryo stained with αsense *Piga* probe (**A, C, E**) or sense *Piga* probe (**B, D, E**). FLAER flow cytometry staining of WT (orange, blue) and *Piga* hemizygous cKO MPNMCs (green). FLAER MFI quantified (**H**). Fb = Forebrain, Mb = Midbrain, BA1 = Branchial Arch 1, MNP = Medial Nasal Process, Lb = Limb bud. **p<0.01.

DOI: https://doi.org/10.7554/eLife.45248.022

The following source data and figure supplement are available for figure 8:

**Source data 1.** MPNMC FLAER MFI.

DOI: https://doi.org/10.7554/eLife.45248.024

**Figure supplement 1.** GPI biosynthesis genes show increased expression in the first branchial arch, limb bud, and forebrain.

DOI: https://doi.org/10.7554/eLife.45248.023

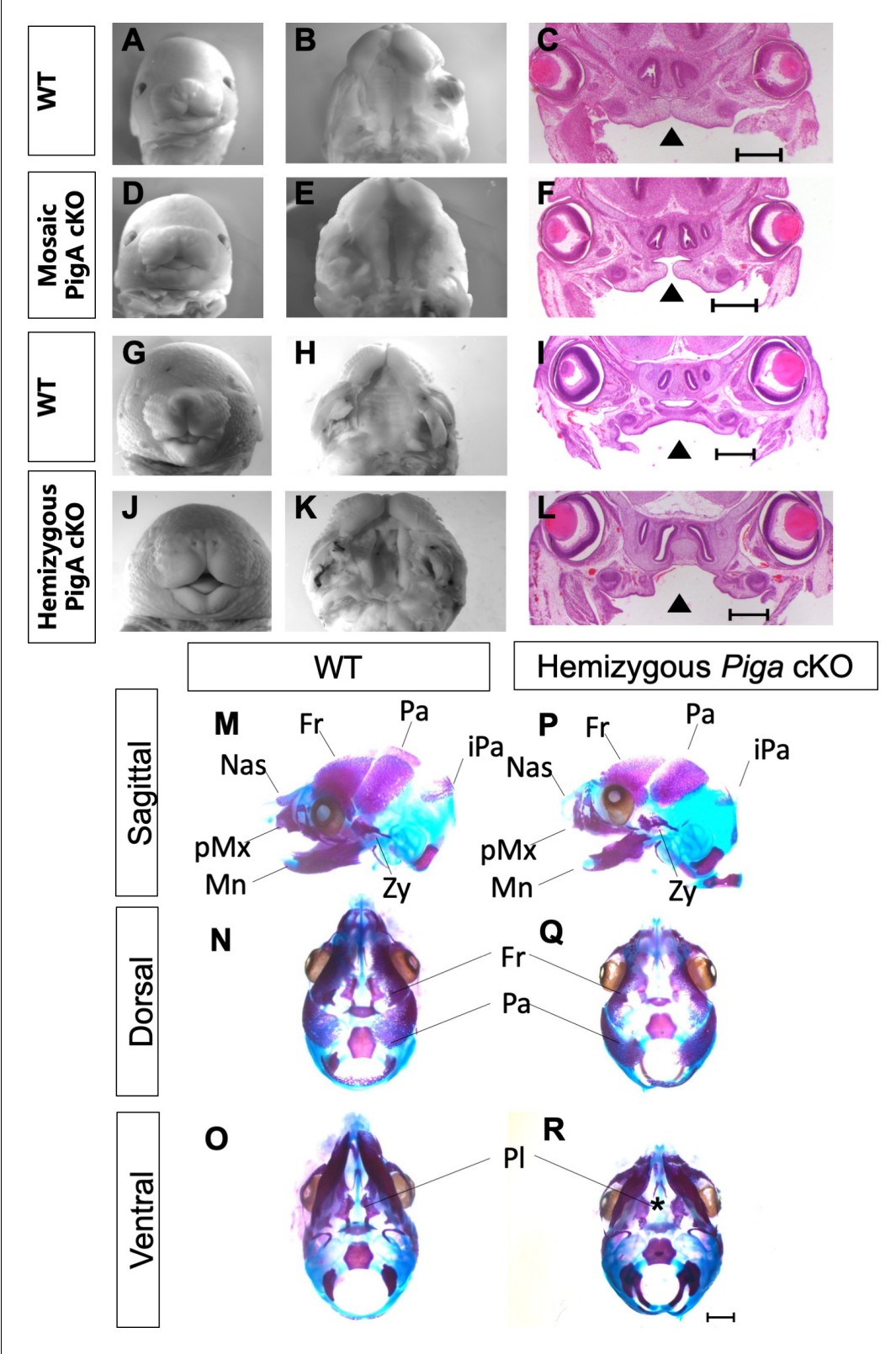

**Figure 9.** Conditional knockout of *Piga* abolishes GPI biosynthesis in NCCs and leads to median cleft lip/palate and craniofacial hypoplasia. Whole mount images of E15.5 WT (**A**), mosaic *Piga* cKO (**D**), E16.5 WT (**G**) and hemizygous *Piga* cKO (**J**). Ventral view of the secondary palate of E15.5 WT (**B**), Mosaic cKO (**E**), E16.5 WT (**H**) and hemizygous cKO (**K**). H and E staining of E15.5 WT (**C**), mosaic cKO (**F**), E16.5 WT (**I**), and hemizygous cKO (**L**), arrowhead indicates cleft palate. Alazarin red and alcian blue staining of E16.5 WT skull (**M–O**) and hemizygous *Piga* cKO skull (**P–R**). Asterick indicates

*Figure 9 continued on next page*

*Figure 9 continued*

cleft palate. Fr = Frontal bone, Pa = Parietal bone, iPa = interparietal bone, Zy = Zygomatic bone, Mn = Mandible, pMx = Premaxilla, Nas = Nasal bone. Scale bar indicates 500 μM in C, F, I, L and 1 mm in M-R.

DOI: https://doi.org/10.7554/eLife.45248.025

aspect of the nasal pits during lip closure at E9.5-E11.5. In later stages of organogenesis, *Pgap2* is widely expressed and enriched in epithelia. These data argue expression of GPI biosynthesis genes is dynamic during development and not simply uniform and ubiquitous. FLAER flow cytometry and expression of a tagged FOLR1 showed that reduced levels of *Pgap2* affected GPI biosynthesis, although not as severely as a total knockout for the GPI biosynthesis pathway, *PIGA*$^{null/null}$. Molecular analysis showed the *Clpex* mutants have increased apoptosis in cNCCs and cranial neuroepithelium. Folinic acid diet supplementation in utero partially rescued the cleft lip in *Clpex* mutants. Finally, we generated a NCC tissue-specific GPI-deficient model to determine the cell autonomous role of GPI biosynthesis. *Piga*$^{flox/X}$; *Wnt1*-Cre mosaic cKO mutants and *Piga*$^{flox/Y}$; *Wnt1*-Cre hemizygous cKO mutants displayed fully penetrant median cleft lip/cleft palate and craniofacial hypoplasia similar to our germline *Clpex* mutant, confirming a cell autonomous role for GPI biosynthesis in craniofacial development.

Contrary to previous studies of other GPI biosynthesis pathway genes, we found *Pgap2* clearly shows enriched expression in certain tissues during certain stages of development. We observed a similar pattern in *Piga* RNA expression suggesting GPI biosynthesis genes share similar gene enrichment domains. These tissues are the most affected in GPI biosynthesis mouse mutants and include the craniofacial complex, CNS, limb, and heart. This may mean *Pgap2* and other GPI biosynthesis genes are required in certain tissues for anchoring GPI-APs critical to that tissue. Alternatively, these areas may be particularly 'GPI-rich.'

A variety of mutants have been described in the GPI biosynthesis pathway with a wide array of phenotypes (*Kinoshita, 2014*; *Bellai-Dussault et al., 2019*). While germline mutants in this pathway remain poorly understood, recent research in Paroxysmal Nocturnal Hemoglobinuria (PNH) caused by somatic mutations in *PIGA* has revolutionized our understanding of GPI deficiency related pathology. In PNH, clones of GPI deficient hematopoietic stem cells proliferate in the bone marrow and give rise to blood cells that lack GPI-anchored CD55/59 which are required to prevent complement-mediated lysis of red blood cells. PNH patients suffer from episodes of hemolysis and thrombosis which can be deadly (*Hill et al., 2017*). Blockade of complement in these patients via eculizumab, a monoclonal antibody that inhibits the conversion of C5 to C5a and C5b, has been shown to greatly improve survival (*Brodsky et al., 2008*; *Rother et al., 2007*; *Hillmen et al., 2007*; *Hillmen et al., 2006*). Thus, a single GPI-AP seems to be largely responsible for the disease observed in these patients.

In this study, we aimed to identify a single GPI-AP that could be responsible for all the phenotypes observed in our germline GPI biosynthesis *Clpex* mutant. Of the known GPI-AP knockout models, *Clpex* shares the most phenotypic overlap with the *Folr1*$^{null/null}$ mouse. We directly tested the hypothesis that FOLR1 deficiency is solely responsible for the *Clpex* phenotype by dietary supplementation of folinic acid during embryonic development. To our surprise, folinic acid supplementation could partially rescue the cleft lip phenotype, but not the NTD or cleft palate (*Figure 6B,C*). Folinic acid treatment could not rescue the apoptosis we observed in control diet treated *Clpex* mutants suggesting folinic acid is necessary for some other aspect of lip development in *Clpex* mutants (*Figure 7—figure supplement 1*). The failure of folinic acid to rescue the apoptosis in the cNCCs and in the neuroepithelium of *Clpex* mutants suggests that either, other GPI-APs are responsible for this apoptosis, or a combination of GPI-APs are required to prevent this apoptosis and folinic acid alone is not sufficient to prevent the apoptosis. Alternatively, the mistrafficking of GPI-APs may result in a cellular response such as the unfolded protein response which may trigger apoptosis in these cells. Further research is required to determine whether this may be the case.

The partial rescue of cleft lip in *Clpex* mutants with high doses of folinic acid *in utero* suggests folinic acid may be a possible therapeutic for some phenotypes in patients with GPI biosynthesis variants. Further research is required to test whether the positive effect of folinic acid on the *Clpex* mutants could be observed in other GPI biosynthesis mutants. These data also argue the

phenotypes observed in germline *Clpex* mutants do not share a single mechanism and are not due to the loss of a single GPI-AP given the varied response in different tissues to the rescue regimens utilized here.

Many GPI-APs could be responsible for the phenotypes we observe in the *Clpex* mutant but were not tested explicitly in this work. Notably, the two receptors for Glial Derived Neurotrophic Factor (GDNF) are GPI- anchored (GFRA1, GFRA2). GFRA1 and GFRA2 are known to be critical for the survival and development of NCCs in the gut during enteric nervous system development (*Tomac et al., 2000*; *Enomoto et al., 1998*; *Rossi et al., 1999*). Interestingly, *Gfra1* and *Gfra2* are expressed in the craniofacial complex during development (*Golden et al., 1998*; *Visel, 2004*). Whether GDNF plays a crucial in cNCC survival remains to be explored. Other candidate GPI-APs that may be affected by loss of *Pgap2* including one form of Neural Cell Adhesion Molecule (NCAM), a critical neural cell adhesion molecule. *Ncam1*$^{null/null}$ mice display defects in neural tube development including kinking and delayed closure (*Rabinowitz et al., 1996*). A third candidate includes the glypican family members which are GPI anchored heparin sulfate proteoglycans that play critical roles in cell-cell signaling and have been shown to modulate critical patterning gradients in the neural tube and face including Sonic Hedgehog and Wnt (*Bassuk et al., 2013*; *Capurro et al., 2008*; *Galli et al., 2003*; *Song and Filmus, 2002*). Other GPI-AP knockout models display NTDs including *Repulsive guidance molecule A/B* (*Rgma*) and *Ephrin A5 (Efna5)*. However, *Rgma*$^{null/null}$ mice do not develop increased apoptosis in the neuroepithelium as we observed in *Clpex* mutants (*Niederkofler et al., 2004*). *Efna5*$^{-/-}$ mice appear to form DLHPs, though the neural folds do not fuse in the midline which is less severe than the defect we observe in *Clpex* mutants (*Holmberg et al., 2000*). Therefore, we find it unlikely the loss of these GPI-APs are primarily responsible for the defects observed in the *Clpex* mutant, although contributions to the phenotype may come from abnormal presentation of one or several of these GPI-APs on the cellular membranes.

It has been known for decades that treatment of embryos with phospholipase C to release GPI-APs from the cell surface causes NTD *in utero* (*O'Shea and Kaufman, 1980*). To investigate the cause of the NTD in *Clpex* mutants, we performed histological and immunohistochemical analysis of the mutant at neurulation stages. We found the *Clpex* mutant fails to form dorsolateral hinge points and the cranial neuroepithelium is apoptotic in the region of the developing DLHP. Neuroepithelial apoptosis was restricted to the midbrain/hindbrain boundary and likely explains why *Clpex* mutants develop cranial NTDs as opposed to caudal NTDs such as spina bifida. These cellular defects likely underlie the NTD but the cause of the neuroepithelial apoptosis remains unclear as the NTD did not respond to folinic acid supplementation. It remains controversial, but the NTD in *Folr1*$^{null/null}$ mice may be related to an expansion of the *Shh* signaling domain that patterns the neural tube (*Tang and Finnell, 2003*; *Murdoch and Copp, 2010*). Indeed, many *Shh* gain-of-function mutants develop NTD as *Shh* expansion impairs the formation of DLHPs and closure of the neural tube (*Murdoch and Copp, 2010*). Our RNA sequencing analysis did not identify a dysregulation in the *Shh* signaling pathway so there are likely differences in the mechanism responsible for the NTD in *Folr1*$^{null/null}$ mice and *Clpex* mutants.

To determine alternative mechanisms responsible for the *Clpex* phenotype, we performed RNA sequencing from E9.5 WT and *Clpex* mutants. We found the largest differences in gene expression were in A/P patterning genes and mesendoderm induction genes. The A/P axis and induction of mesendoderm has been shown to require GPI-anchored CRIPTO, a Tgfβ superfamily member co-receptor of NODAL. A variety of studies have shown CRIPTO/Tgfβ super family members pathway function is impaired in GPI biosynthesis mutants because CRIPTO is GPI-anchored and cleavage of the anchor affects CRIPTO function (*McKean and Niswander, 2012*; *Chen et al., 2008*; *Lee et al., 2016*).

While GPI deficiency has been studied in the context of A/P patterning, this is the first study to implicate GPI biosynthesis in the survival of neural crest cells in a cell autonomous fashion. Indeed, the enrichment of *Piga* in the developing medial nasal process and the median cleft lip/cleft palate and craniofacial hypoplasia in our *Piga* cKOs confirms a unique cell autonomous role for GPI biosynthesis in these structures. Interestingly, these mutants do not show a complete loss of the craniofacial skeleton, rather a general, mild hypoplasia consistent with a role for GPI biosynthesis in early NCC survival, but not later patterning or differentiation.

Our study provides potential mechanistic explanations for the developmental defects observed in a GPI biosynthesis mutant model. We propose GPI biosynthesis is involved in anchoring critical survival factors for NCCs and the neuroepithelium. In GPI-deficient states, NCCs undergo apoptosis leading to hypoplastic nasal processes and palatal shelves. As we reduced the degree of GPI biosynthesis from the germline *Clpex* mutant hypomorph to our totally GPI-deficient NCC cKO model, we observed a worsening of the craniofacial phenotype as witnessed by the fully penetrant cleft lip/cleft palate and craniofacial hypoplasia. These data argue the degree of GPI deficiency correlates with the severity of the phenotype. In the neuroepithelium, loss of neuroepithelial cells at the DLHPs result in failure to bend and close the neural tube. Conditional ablations of critical GPI biosynthesis genes in other affected tissues including the CNS and heart will likely lead to new understandings of the diverse pathology of inherited glycophosphatidylinositol deficiency.

# Materials and methods

## Key resources table

| Reagent type (species) or resource | Designation | Source or reference | Identifiers | Additional information |
|---|---|---|---|---|
| Genetic reagent (*Mus Musculus*) | *Piga^flox* | Riken | Riken:B6.129-*Piga^tm1* RRID:IMSR_RBRC06211 | |
| Genetic reagent (*Mus Musculus*) | Wnt1-Cre | Jackson Laboratories | JAX:B6.Cg-*H2afv Tg(Wnt1-cre)11RthTg(Wnt1-GAL4)11Rth*/J RRID:IMSR_JAX:003829 | |
| Genetic reagent (*Mus Musculus*) | R26R LacZ reporter | Jackson Laboratories | JAX:B6.129S4 *Gt (ROSA)26Sor^tm1Sor*/J; R26R^Tg RRID:MGI:2176735 | |
| Genetic reagent (*Mus Musculus*) | *Pgap2^null* | EUCOMM | EUCCOM: *Pgap2 tm1a(EUCOMM)Wtsi* RRID:IMSR_EM:09276 | |
| Genetic reagent (*Mus Musculus*) | *Clpex* | *Stottmann et al., 2011* | In house: *Pgap2^Clpex* RRID:MGI:5056383 | |
| Cell line (*Homo sapiens*, kidney) | HEK 293 T cell | ATCC | ATCC: #CRL-11268 RRID:CVCL_1926 | |
| Antibody | sheep polyclonal anti-Digoxigenin | Roche | Roche: #11093274910 RRID:AB_2734716 | (1:5000) |
| Antibody | rabbit polyclonal anti-myc | Abcam | Abcam: #ab9106 RRID:AB_307014 | (1:1000) |
| Antibody | 488-congugated goat polyclonal anti-rabbit | Thermo | Thermo: #A11008 RRID:AB_143165 | (1:500) |
| Antibody | mouse monoclonal anti-AP2 | Developmental Studies Hybridoma Bank | DSHB: #3B5 supernatant RRID:AB_528084 | (1:20) |
| Antibody | rabbit polyclonal anti-Cleaved Caspase 3 | Cell Signaling Technology | CST: #9661 RRID:AB_2341188 | (1:300) |
| Antibody | Mouse monoclonal anti-myc | Sigma | Sigma: #M4439-100UL RRID:AB_439694 | (1:2000) |
| Antibody | Mouse monoclonal anti-Tubulin | Sigma | Sigma: #T6199 RRID:AB_477583 | (1:1000) |
| Antibody | goat anti-rabbit IRDye 800CW | LICOR | LICOR: # 926–32211 RRID:AB_621843 | (1:15000) |
| Antibody | goat anti-mouse IRDye 680Rd | LICOR | LICOR: #926–68070 RRID:AB_10956588 | (1:15000) |
| Sequence-based reagent | paired-end RNA sequencing | Beijing Genomics Institute-Americas | | |

*Continued on next page*

*Continued*

| Reagent type (species) or resource | Designation | Source or reference | Identifiers | Additional information |
|---|---|---|---|---|
| Peptide, recombinant protein | BBSI enzyme | New England Biolabs | NEB: R0539S | |
| Commercial assay or kit | MEGAclear Transcription Clean-up kit | Thermo | Thermo: #AM1908 | |
| Chemical compound, drug | Alexafluor-488 proaerolysin (FLAER) | CedarLane Labs, Burlington, Ontario | Cedarlane Labs: #FL1-C, 25 µg | |
| Chemical compound, drug | 25ppm folic acid diet | Envigo | Envigo: Custom diet TD.160472 | |
| Chemical compound, drug | 25ppm folinic acid diet | Envigo | Envigo: Custom diet TD.160746 | |
| Chemical compound, drug | Control Diet | Envigo | Envigo: Custom diet TD.160112 | |
| Chemical compound, drug | 5-Bromo-4-chloro-3-indolyl β-D-galactopyranoside | Sigma | Sigma: #B4252 | |
| Chemical compound, drug | Alcian Blue | Sigma | Sigma: #A3157 | |
| Chemical compound, drug | Alazarin Red | Sigma | Sigma: #A5533 | |
| Chemical compound, drug | Toluidine Blue | Sigma | Sigma: #89640 | |
| Chemical compound, drug | wheat germ agglutinin Texas Red Conjugate | Thermo | Thermo: #W21405 | (5 µL WGA/1 mL PBS) |
| Chemical compound, drug | Folinic acid | Sigma | Sigma: #F7878-500MG | |
| Software, algorithm | Graphpad Prism | GraphPad Software, San Diego, CA | RRID:SCR_002798 | |
| Software, algorithm | Imaris 9.2.1, colocalization function | Oxford Instruments | RRID:SCR_007370 | |
| Software, algorithm | Nikon Elements Software, birghtspot analysis | Nikon Instruments Inc. | RRID:SCR_014329 | |
| Software, algorithm | FASTQC | https://www.bioinformatics.babraham.ac.uk/projects/fastqc/ | RRID:SCR_014583 | |
| Software, algorithm | RSEM-v1.3.0 | *Li and Dewey, 2011* | RRID:SCR_013027 | |
| Software, algorithm | Toppgene | https://toppgene.cchmc.org/ | RRID:SCR_005726 | |
| Software, algorithm | Computational Suite for Bioinformaticians and Biologists | https://github.com/csbbcompbio/CSBB-v3.0 | RRID:SCR_017234 | |
| Software, algorithm | Benchling sgRNA design software | Benchling, San Fransisco CA | RRID:SCR_013955 | |
| Recombinant DNA reagent | mouse Piga plasmid | Origene | Origene: #MR222212 | |
| Recombinant DNA reagent | mouse Pgap2 plasmid | Origene | Origene: #MR2031890 | |
| Recombinant DNA reagent | mouse Pigp plasmid | Origene | Origene: #MR216742 | |
| Recombinant DNA reagent | mouse Pigu plasmid | Origene | Origene: #MR223670 | |

*Continued on next page*

*Continued*

| Reagent type (species) or resource | Designation | Source or reference | Identifiers | Additional information |
|---|---|---|---|---|
| Recombinant DNA reagent | mouse Pigx plasmid | Origene | Origene: #MR201059 | |
| Recombinant DNA reagent | mouse Lhx8 plasmid | Origene | Origene: #MR226908 | |
| Recombinant DNA reagent | mouse Tbxt plasmid | Origene | Origene: #MR223752 | |
| Recombinant DNA reagent | mouse Alx3 plasmid | DNASU | DNASU: #MmCD00081160 | |
| Recombinant DNA reagent | CRISPR/Cas9 PX459M2 puromycin-resistance vector | *Ran et al., 2013* | | |
| Recombinant DNA reagent | Ultramer with 5' and 3' phosphorothiolate bonds | Integrated DNA Technologies | | |
| Transfected construct (*Homo sapiens*) | Human N-myc tagged Folr1 plasmid | Sinobiological | Sinobiological: #HG11241-NM | |

## Animal husbandry

All animals were maintained through a protocol approved by the Cincinnati Children's Hospital Medical Center IACUC committee (IACUC2016-0098). Mice were housed in a vivarium with a 12 hr light cycle with food and water *ad libitum*. The Clpex line was previously published by *Stottmann et al. (2011)*. *Piga$^{flox}$* (B6.129-*Piga$^{tm1}$*) mice were obtained from RIKEN and were previously generated by Taroh Kinoshita and Junji Takeda (*Nozaki et al., 1999*). *Wnt1*-Cre (B6.Cg-*H2afv$^{Tg(Wnt1-cre)11Rth}$*Tg$^{(Wnt1-GAL4)11Rth}$/J) mice and R26R LacZ reporter (B6.129S4 *Gt(ROSA)26Sor$^{tm1Sor}$*/J; R26R$^{Tg}$) mice were purchased from Jackson Laboratories and previously published. *Pgap2$^{null}$* (*Pgap2$^{tm1a(EUCOMM)Wtsi}$*) mice were obtained from EUCOMM and genotyped using their suggested primers. Primers used to genotype all animals are listed in *Supplementary file 2*. Sample2SNP custom Taqman probes were designed by Thermo-Fisher and used to genotype the point mutation in the *Clpex* line.

## Mapping and sequencing

Mapping of the *Clpex* mutation was previously described (*Stottmann et al., 2011*). Whole exome sequencing was done at the CCHMC DNA Sequencing and Genotyping Core. The *Pgap2* exon three variant was Sanger sequenced after PCR amplification and purification using the Zymo DNA clean and Concentrator kit (Zymo Research Corporation, Irvine, CA).

## Alternative transcript analysis

Alternative transcripts of *Pgap2* were identified using Ensembl Genome browser and the UCSC genome browser (*Hunt et al., 2018*; *Kent et al., 2002*). Transcripts were aligned using ExPasy software (Swiss Institute of Bioinformatics, Switzerland) (*Gasteiger et al., 2003*).

## Whole mount in situ hybridization

RNA *in situ* hybridization was performed as previously described (*Belo et al., 1997*). Briefly, whole E8-E11.5 embryos were fixed overnight in 4% PFA at 4°C and dehydrated through a methanol series. Samples were treated with 4.5 µg/mL Proteinase K for 7–13 min at room temperature, post-fixed in 4% PFA/0.2% glutaraldehyde and blocked with hybridization buffer prior to hybridization overnight at 65°C with constant agitation. The samples were washed and incubated with an anti-Digoxigenin antibody (Roche #11093274910) o/n at 4°C. Embryos were washed and incubated with NBT/BCIP (SIGMA) or BM Purple (Roche #11442074001) from 4 hr at room temperature to o/n at 4°C.

Piga (#MR222212), Pgap2 (#MR2031890) Pigp (#MR216742), Pigu (#MR223670), Pigx (#MR201059), Lhx8 (#MR226908), and Tbxt (#MR223752) plasmids were obtained from Origene

(Rockville, MD). Antisense probes were generated from PCR products containing T3 polymerase overhangs. *Piga*, *Pgap2*, *Pigp*, *Pigu*, *Pigx*, *Lhx8*, and *Tbxt* antisense probes were generated from 910, 750, 556, 952, and 416, 519, and 665 base pair products, respectively. The PCR products were purified, in vitro transcription was performed with digoxigenin-labeled dUTP (Roche #11277073910), and the probe was purified with the MEGAclear Transcription Clean-up kit (Thermo #AM1908) per the manufacturer's instructions. For sense probes, the plasmids were cut with XhoI restriction enzyme after the coding sequence and T7 RNA polymerase was used for in vitro transcription. The *Alx3* probe was generated by in vitro transcription of a 790 bp PCR product from *Alx3* plasmid (DNASU #MmCD00081160) containing T3 polymerase overhangs.

## MEF/MPNMC production and FLAER staining

MEFs were generated from E13.5 embryos. Embryos were dissected in PBS, decapitated, and eviscerated. The remaining tissue was incubated in trypsin o/n at 4°C to allow for enzymatic action on the tissue and remaining fibroblasts were passaged in complete DMEM containing 10%FBS and penicillin/streptomycin. MEFs were stained within three passages of their isolation. MEFs and 293 T cells were stained with 5 µL of Alexafluor-488 proaerolysin (FLAER)/$1 \times 10^6$ cells (CedarLane Labs, Burlington, Ontario, Canada) and flow cytometry was performed on Becton-Dickinson FACSCanto II flow cytometer in the CCHMC Research flow cytometry core. Mouse Palatal Nasal Mesenchymal Cells (MPNMCs) were generated from E13.5-E14.5 microdissected embryo heads in a protocol similar to that used for MEPMS (*Fantauzzo and Soriano, 2017*). The lower jaw, eyes and brain were removed and the remaining upper jaw and nasal mesenchyme were lysed in 0.25% trypsin for 10 min at 37°C, passaged through a P1000 pipette several times to create a single-cell suspension, and cultured in 12 well plates. These cells displayed a stellate mesenchymal cell appearance after culture overnight. They were then stained 72 hr after isolation with FLAER.

## CRISPR knockout/knock-in gene editing

We utilized a double guide approach to generate knockout clones with deletions in *PGAP2* and *PIGA* in HEK293T cells. Two small guide RNAs targeting exon 3 of either *PGAP2* or *PIGA* were designed using Benchling software (Benchling, San Francisco, CA) and 5' overhangs were added for cloning into CRISPR/Cas9 PX459M2 puromycin-resistance vector (*Ran et al., 2013*). We also generated a single gRNA and donor oligonucleotide for homologous recombination to recapitulate the *Clpex* mutation in 293 T cells (Integrated DNA Technologies ultramer). We cloned these guides into the PX459M2 plasmid using the one-step digestion-ligation with BbsI enzyme as described by Ran et. al. (*Ran et al., 2013*). For knockout line, two guides per gene were transfected in WT 293 T cells using Lipofectamine 3000. For *Clpex* knock-in lines, one sgRNA and the donor oligonucleotide containing 5' and 3' phosphorothiolate bonds were transfected into WT 293 T cells using Lipofectamine 3000. Cells were selected for transfection by 3 days of culture in 10 µg/mL puromycin. Transfected cells were plated at clonal density into a 96-well plate and single clones were scored approximately one week post seeding. Single clones were Sanger sequenced to confirm deletion of the target exon 3 sequence of either *PIGA* or *PGAP2*. *PIGA* clones carry a 50 bp out-of-frame deletion in *PIGA* and lack virtually all GPI expression on the cell surface by FLAER flow cytometry staining. *PGAP2* clones carry a 121 bp out-of-frame deletion in *PGAP2*. Clpex Knock-in clones were Sanger sequenced to identify clones carrying the desired knock-in mutation and clones with indels were discarded. Primers for sgRNA cloning and PCR amplification of targeted regions can be found in *Supplementary file 2*. Sequencing of clones is presented in *Figure 4—figure supplement 1*. HEK293T cells were purchased from ATCC and STR profiling at Genetica LabCorp. showed a HEK293 match. Cells were found to be mycoplasma negative via a LookOut Mycoplasma PCR Detection Kit (SIGMA).

## Immunofluorescence

293 T cells were transfected with FOLR1-myc constructs (Origene #RC212291) using Lipofectamine 3000, incubated for 48 hr, then fixed for 15 min in 4% PFA, and blocked in 4% normal goat serum in PBS. They were stained o/n at 4°C with 1:1000 rabbit anti-myc (Abcam ab9106), washed the next day and stained with 1:500 488-conjugated goat anti-rabbit (Thermo #A11008). They were then stained for 5 min with 5 µg/mL wheat germ agglutinin (Thermo Fischer #W21405) and counter-

stained with DAPI. They were visualized on Nikon C2 confocal microscope and Pearson co-efficient analysis was performed by Imaris software utilizing 8–15 60x z-stack confocal images/genotype.

E9.5 embryos were dissected, fixed in 4% PFA o/n, equilibrated in 30% sucrose o/n, cryo-embedded in OCT, and sectioned from 10 to 20 µM by cryostat. Sections were subjected to antigen retrieval by citrate retrieval buffer, blocked in 4% normal goat serum, incubated in primary antibody 1:20 mouse anti-AP2 (Developmental Studies Hybridoma Bank, University of Iowa, 3B5 supernatant) and 1:300 rabbit anti-Cleaved Caspase 3 (Cell Signaling Technology, Danvers, MA #9661) o/n in humid chamber. Sections were incubated with secondary antibody 1:500 Alexafluor 488-conguated goat anti-rabbit (Thermo #A11008) and 1:500 Alexafluor 594 conjugated goat anti-mouse (Thermo A11008) and counterstained with DAPI. Sections were imaged on Nikon C2 confocal microscope and CC3+ cells and AP2+ cells were quantified with Nikon Elements software brightspot analysis.

## Western blotting

293 T cells were transfected with FOLR1-myc constructs using Lipofectamine 3000, incubated for 48 hr and lysed in Pierce RIPA buffer (Thermo #89901) containing Protease Inhibitor cocktail (Roche #11697498001). Lysate protein concentration was determined by BCA assay and electrophoresis was performed on a 10% Tris-glycine gel. Protein was transferred to a PVDF membrane, blocked in Odyssey blocking buffer and incubated o/n at 4°C with 1:1000 Rabbit anti-myc (Abcam ab9106) and 1:1000 Mouse anti-Tubulin (Sigma #T6199) antibodies). Membranes were washed and incubated for 1 hr in 1:15000 goat anti-rabbit IRDye 800CW (LICOR # 926–32211)and 1:15000 goat anti-mouse IRDye 680Rd (LICOR, #926–68070) and visualized on LICOR Odyssey imaging system.

## Histology

Whole embryos E8-E16.5 were fixed in formalin and embedded in paraffin for coronal sectioning and stained with hematoxylin and eosin using standard methods.

## NCC lineage trace and Xgal staining

*Clpex* heterozygous females were crossed to *Wnt1*-Cre R26R transgenic mice as described in results. Whole embryos were fixed in 4% PFA for 15 min at RT, washed in lacZ buffer, and stained in a solution containing 1 mg/mL X-gal (Sigma #B4252) (*Behringer, 2014*). They were washed three times in PBS-T and imaged after several hours in X-gal stain at room temperature.

## Diet

*Clpex* pregnant dams were treated with either control chow, chow +25 ppm folic acid, or chow +25 ppm folinic acid generated by Envigo (Indianapolis, Indiana) from E0-E16.5 *ad libitum* (Control diet #TD.160112, Folic acid diet #TD.160472, Folinic acid diet #TD.160746). They were euthanized at either E9.5 or E16.5 to assess phenotype.

## RNA sequencing

5 WT and 5 *Clpex* mutant E9.5 embryos were dissected from the yolk sac and snap frozen on dry ice. RNA was isolated and pooled samples of each genotype were used for paired-end bulk-RNA sequencing (BGI-Americas, Cambridge, MA). RNA-Seq analysis pipeline steps were performed using CSBB [Computational Suite for Bioinformaticians and Biologists: https://github.com/csbbcompbio/CSBB-v3.0]. CSBB has multiple modules, RNA-Seq module is focused on carrying out analysis steps on sequencing data, which comprises of quality check, alignment, quantification and generating mapped read visualization files. Quality check of the sequencing reads was performed using FASTQC (http://www.bioinformatics.bbsrc.ac.uk/projects/fastqc). RNA-Seq reads for the mutant and wildtype were paired-end and had ~43 and~31 million reads respectively. Reads were mapped (to mm10 version of Mouse genome) and quantified using RSEM-v1.3.0 (*Li and Dewey, 2011*). Differential expression analysis was carried out by EBSeq [https://www.biostat.wisc.edu/~kendzior/EBSEQ/] (*Leng et al., 2013*). Differential transcripts are filtered based on LogFC and p-value. Filtered DE transcripts are used for functional and pathway enrichment using toppgene [https://toppgene.cchmc.org/] (*Chen et al., 2009*).

## Skeletal preparation

For skeletal preparation, E16.5-E18.5 embryos were eviscerated and fixed for 2 days in 95% ethanol. They were stained overnight at room temperature in Alcian blue solution (Sigma #A3157) containing 20% glacial acetic acid. They were destained for 24 hr in 95% ethanol and slightly cleared in a 1% KOH solution o/n at room temperature. They were then stained o/n in Alazarin red solution (Sigma #A5533) containing 1% KOH. They were then cleared for 24 hr in 20% glycerol/1%KOH solution. Finally, they were transferred to 50% glycerol/50% ethanol for photographing.

## Barrier function assay

E18.5 embryos were dehydrated through a methanol series and then rehydrated. Next, they were placed in 0.1% Toludine Blue (Sigma #89640) in water for 2 min on ice. They were destained in PBS on ice and imaged.

## Statistical analysis

Statistical analysis was performed using Graphpad Prism (GraphPad Software, San Diego, CA). Tests between two groups were carried out using unpaired, two-tailed student's t-test. Diet, comparisons between groups for FLAER staining, and subcellular localization (Pearson Correlation co-efficient) results were analyzed with one-way ANOVA on Graphpad Prism (Graphpad Software, San Diego, CA). For statistical analysis of phenotypes observed for embryos under varying diet conditions, z-test of proportions was used. Significance was labeled with one asterisk = $p < 0.05$, two asterisks = $p < 0.01$, three asterisks = $p < 0.001$, and four asterisks = $p < 0.0001$.

# Acknowledgements

This work was supported by the Cincinnati Children's Research Foundation, NIH (RWS R01NS085023) and the American Cleft Palate -Craniofacial Association (Paul W Black, MD Grant for Emerging Researchers to MJL).

# Additional information

### Funding

| Funder | Grant reference number | Author |
| --- | --- | --- |
| National Institute of Neurological Disorders and Stroke | R01NS085023 | Rolf W Stottmann |
| American Cleft Palate-Craniofacial Association | | Marshall Lukacs |

The funders had no role in study design, data collection and interpretation, or the decision to submit the work for publication.

### Author contributions

Marshall Lukacs, Conceptualization, Data curation, Formal analysis, Funding acquisition, Investigation, Methodology, Writing—original draft, Writing—review and editing; Tia Roberts, Methodology; Praneet Chatuverdi, Formal analysis; Rolf W Stottmann, Conceptualization, Data curation, Formal analysis, Supervision, Funding acquisition, Writing—original draft, Project administration, Writing—review and editing

### Author ORCIDs

Rolf W Stottmann (iD) https://orcid.org/0000-0003-4512-6806

### Ethics

Animal experimentation: This study was performed in strict accordance with the recommendations in the Guide for the Care and Use of Laboratory Animals of the National Institutes of Health. All of the animals were handled according to approved institutional animal care and use committee

(IACUC) protocol #2016-0098 of the Cincinnati Children's Hospital Medical Center. All euthanasia was performed after isoflourane sedation, and every effort was made to minimize suffering.

## Decision letter and Author response
Decision letter https://doi.org/10.7554/eLife.45248.036
Author response https://doi.org/10.7554/eLife.45248.037

## Additional files

### Supplementary files
• Supplementary file 1. *Pgap2* alternatively spliced transcripts.
DOI: https://doi.org/10.7554/eLife.45248.026

• Supplementary file 2. Primers used in this study.
DOI: https://doi.org/10.7554/eLife.45248.027

• Supplementary file 3. *Clpex* mutants display defects in expression of anterior/posterior patterning genes. E9.5 WT (A,C) and *Clpex* mutant (B, D) RNA in situ hybridization with α-sense *Alx3* probe, an anterior pattering gene. E9.5 WT (E) and *Clpex* mutant (F) RNA in situ hybridization with α-sense *Lhx8* probe, an anterior patterning gene. E9.5 WT (G) and *Clpex* mutant (H) RNA in situ hybridization with α-sense *Tbxt* (*Brachyury*) probe, a posterior patterning gene. E8.5 WT (I) and *Clpex* mutant (J) RNA in situ hybridization with α-sense *Tbxt* (*Brachyury*) probe, a posterior patterning gene.
DOI: https://doi.org/10.7554/eLife.45248.028

• Transparent reporting form
DOI: https://doi.org/10.7554/eLife.45248.029

### Data availability
Sequencing data have been submitted to GEO (Clpex exome seq: GSE131920 and RNA-Seq: GSE131919).

The following datasets were generated:

| Author(s) | Year | Dataset title | Dataset URL | Database and Identifier |
|---|---|---|---|---|
| Lukacs M, Chatuverdi P, Stottmann R | 2019 | Transcriptome profiling of Embryonic day 9.5 WT and Clpex mutant mice | https://www.ncbi.nlm.nih.gov/geo/query/acc.cgi?acc=GSE131919 | NCBI Gene Expression Omnibus, GSE131919 |
| Lukacs M, Chatuverdi P, Stottmann R | 2019 | Exome sequencing identified mutations in clpex mouse mutants | https://www.ncbi.nlm.nih.gov/geo/query/acc.cgi?acc=GSE131920 | NCBI Gene Expression Omnibus, GSE131920 |

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
