## [Decision Letter]

Thank you for submitting your article "Glycosylphosphatidylinositol biosynthesis and remodeling are required for neural crest, cardiac and neural development" for consideration by *eLife*. Your article has been reviewed by three peer reviewers, and the evaluation has been overseen by Joseph Gleeson as the Reviewing Editor and Marianne Bronner as the Senior Editor. The following individual involved in review of your submission has agreed to reveal his identity: Richard H Finnell (Reviewer #2).

The reviewers have discussed the reviews with one another and the Reviewing Editor has drafted this decision to help you prepare a revised submission.

Summary:

Lukacs et al. describe the cloning and characterization of a novel ENU mutant, focusing on the finding that the GPI anchoring biosynthetic process is required for neural crest cell and craniofacial development. The authors clearly demonstrate that Pgap2 is the mutated gene in the Clpex mutant, and convincingly show that GPI- anchoring is disrupted in these mutants. The authors also make some interesting observations with regard to the tissue-specific functions of a pathway that one would expect to be more or less housekeeping, highlighting the novelty of their hypomorphic allele. Interestingly, some of the phenotypes can be rescued with folinic acid, bypassing the GPI-anchored Folr1 and Folr2. Overall, the manuscript is an interesting addition to our body of knowledge concerning the GPI biosynthetic pathway and neural crest development. Pgap2 is involved in maturation of GPI anchors: particularly Pgap2 is necessary for reacylation in fatty acid remodeling, in which unsaturated fatty acid at the sn2 position of π is replaced by stearic acid. Under Pgap2 defective conditions, fatty acid remodeling stops at an intermediate that bears only one fatty acid. Such intermediate GPI-APs are anchored to the membrane by single fatty acid and prone to be released from the membrane, resulting in the decreased surface GPI-AP levels.

This manuscript is very well written and focuses on what might be considered an understudied yet important aspect of biology. Given the early lethal nature of null GPI pathway mutants, previous studies have been restricted to gastrulation stage embryos and earlier timepoints. Here the authors were able to use the hypomorphic Clpex allele for Pgap2 to identify tissue-specific requirements for GPI synthesis in late-stage mouse embryos. Additionally, using conditional Piga mutants, they were able to study these processes in a cell type specific manner in the cranial neural crest. Thus, the experiments and the findings are novel and important.

Essential revisions:

1) Figure 4C, D. Authors knocked in (KI) the corresponding mutation into PGAP2 gene in human HEK293T cells and assessed cell surface GPI-APs levels by staining with FLAER. PGAP2 KO cells and PIGA KO cells were also generated and compared with the KI cells. PGAP2 KO greatly reduced GPI-AP levels whereas PIGA KO completely abolished GPI-APs. This is consistent with the previous knowledge that PGAP2 null mutant cells express low levels of GPI-APs. GPI-AP levels significantly reduced in 2 of 3 KI clones as expected, however the levels were similar to that of the PGAP2 KO cells (Figure 4C, D and subsection “Pgap2 is required for the proper anchoring of GPI-APs, including FOLR1”, second paragraph). This result could be interpreted that the base substitution in the initiation codon in human PGAP2 (corresponding to mouse Clepx mutation) is a null mutation, which phenotypically causes a partial defect in GPI-AP levels. The authors conclusion that mouse Clepx mutation is hypomorphic is correct considering that Pgap2 null mice are lethal by E9 in development whereas Clepx mice develops until E18.5. Please clarify this point better in the text.

2) Figure 4E-M. Authors conclude that cell surface expression of FOLR1 is abnormal in PGAP2-/- cells (subsection “Pgap2 is required for the proper anchoring of GPI-APs, including FOLR1”, last paragraph). Due to a rather strong intracellular fluorescence, evaluation of cell surface FOLR1 and co-localization with WGA are difficult (comparison of K and M) and picture of one cell is not sufficient for convincing interpretation. Authors should stain non-permeable, intact cells to avoid intracellular staining. Data should be quantitated. In another direction, surface levels of FOLR1 could be measured by flow cytometry. The effect of intracellular fluorescence can be avoided and quantitative data from many cells are easily obtained in flow cytometry (like Figure 4A-D).

3) Figure 4N. It is not possible to evaluate whether bands are specific for FOLR1 because molecular size markers are not shown and because development with isotype matched control antibody is missing. Considering a lot of intracellular staining shown in Figure 4H-M, the main band might correspond to the ER-form of FOLR1 and weaker bands above the main band represent Golgi and cell surface forms.

4) The authors speculate that alternative start sites and/or transcripts might account for the hypomorphism. Is there evidence for this? Commercial antibodies are available, and thus a Western could be performed to show the remaining isoforms, and whether there is enhancement of their translation in mutants. In addition, the authors should describe which possible downstream starts could be utilized and what the effect might be on the protein based on known functional domains.

---

## [Author Response]

Essential revisions:1) Figure 4C, D. Authors knocked in (KI) the corresponding mutation into PGAP2 gene in human HEK293T cells and assessed cell surface GPI-APs levels by staining with FLAER. PGAP2 KO cells and PIGA KO cells were also generated and compared with the KI cells. PGAP2 KO greatly reduced GPI-AP levels whereas PIGA KO completely abolished GPI-APs. This is consistent with the previous knowledge that PGAP2 null mutant cells express low levels of GPI-APs. GPI-AP levels significantly reduced in 2 of 3 KI clones as expected, however the levels were similar to that of the PGAP2 KO cells (Figure 4C, D and subsection “Pgap2 is required for the proper anchoring of GPI-APs, including FOLR1”, second paragraph). This result could be interpreted that the base substitution in the initiation codon in human PGAP2 (corresponding to mouse Clepx mutation) is a null mutation, which phenotypically causes a partial defect in GPI-AP levels. The authors conclusion that mouse Clepx mutation is hypomorphic is correct considering that Pgap2 null mice are lethal by E9 in development whereas Clepx mice develops until E18.5. Please clarify this point better in the text.

We have edited the first section of the Results section to clarify this point.

2) Figure 4E-M. Authors conclude that cell surface expression of FOLR1 is abnormal in PGAP2-/- cells (subsection “Pgap2 is required for the proper anchoring of GPI-APs, including FOLR1”, last paragraph). Due to a rather strong intracellular fluorescence, evaluation of cell surface FOLR1 and co-localization with WGA are difficult (comparison of K and M) and picture of one cell is not sufficient for convincing interpretation. Authors should stain non-permeable, intact cells to avoid intracellular staining. Data should be quantitated. In another direction, surface levels of FOLR1 could be measured by flow cytometry. The effect of intracellular fluorescence can be avoided and quantitative data from many cells are easily obtained in flow cytometry (like Figure 4A-D).

To address this concern we have repeated the immunocytochemistry without permeabilization with 60x confocal imagining and performed colocalization analysis (with Pearson coefficient quantification) to determine the degree of co-localization between surface wheat germ agglutinin staining and surface myc-tagged FOLR1 staining. This is presented in a new Figure 5 and we have included a field of cells for each genotype so the reader can clearly see the deficiency in surface staining for myc-tagged FOLR1 in both PIGA^-/-^ and PGAP2^-/-^ cells. Consistent with FLAER staining, we saw that PIGA^-/-^ cells have a much lower Pearson coefficient (approximately 0) than WT cells and PGAP2^-/-^ cells which have an intermediate value (~0.2).

3) Figure 4N. It is not possible to evaluate whether bands are specific for FOLR1 because molecular size markers are not shown and because development with isotype matched control antibody is missing. Considering a lot of intracellular staining shown in Figure 4H-M, the main band might correspond to the ER-form of FOLR1 and weaker bands above the main band represent Golgi and cell surface forms.

We have repeated the western blot and shown molecular size markers. We have also probed the same lysate with isotype matched control Rabbit IgG antibody. This is now shown in Figure 5K, L.

4) The authors speculate that alternative start sites and/or transcripts might account for the hypomorphism. Is there evidence for this? Commercial antibodies are available, and thus a Western could be performed to show the remaining isoforms, and whether there is enhancement of their translation in mutants. In addition, the authors should describe which possible downstream starts could be utilized and what the effect might be on the protein based on known functional domains.

To address this comment, we have performed PGAP2 western blotting in a variety of contexts with a commercial polyclonal antibody against Pgap2 available from Abcam and Thermo Fischer (abcam #ab175493, Thermo #PA5-64091) We confirmed that both Abcam and Thermo sell the same polyclonal antibody. We both overexpressed myc-tagged PGAP2 in HEK293T cells and knocked out PGAP2 in 293T cells as positive and negative controls for this experiment. Unfortunately, we were unable to detect PGAP2 at the predicted molecular weight of approximately 29.4 kDa in overexpression constructs (see Author response image 1). We confirmed the overexpression plasmid identified Pgap2 by probing with a rabbit αMyc antibody which detected a protein at approximately 20kDa for Pgap2-myc (Author response image 1) We were likewise unable to detect endogenous PGAP2 in either HEK293T cells or in mouse tissue from WT and Clpex mutants (Author response image 1 and data not shown). We also obtained an antibody against PGAP2 via (generous gift of Yusuke Maeda) but were unable to identify a PGAP2 specific band and the collaborator confirmed that they were unable to detect endogenous PGAP2 with the same antibody. Therefore, we conclude that these antibodies do not detect PGAP2 and we therefore cannot definitely address whether or not alternatively spliced forms of PGAP2 protein are made in vivo in Clpex mutants. We have updated the text of the Results to reflect this finding.

**Author response image 1. respfig1:** Commercial Pgap2 antibody does not detect overexpressed or endogenous PGAP2. 10μg lysate from overexpression of mouse Pgap2-myc tagged clone (Origene #MR203189) in WT (lane 2) or Pgap2 KO 293T (lane 3) or lysate from mock transfection of WT (lane 4) or Pgap2 KO 293T (lane 5) probed with 1:500 Rb αPGAP (Thermo #PA5-64091, Green). (**A**) 10μg lysate from overexpression of mouse Pgap2-myc tagged clone in WT (lane 2) or Pgap2 KO 293T (lane 3) or lysate from mock transfection of WT (lane 4) or Pgap2 KO 293T (lane 5) probed with 1:1000 Rb αMyc (Abcam #ab9106, Green) or Ms αTub (Red) (**B**).

We have included Supplementary File 1 which depicts all alternatively spliced transcripts of Pgap2. According to Ensembl, there are 25 distinct alternatively spliced transcripts, 13 of which are protein coding. We have included Figure 2—figure supplement 1 which contains an alignment of alternatively spliced transcript Pgap2-203 with the canonical transcript Pgap2-225, showing the difference in the C-terminal domains between the transcript variants caused by utilization of alternative promoters. We discuss the differences in features between the variants in the manuscript.